# Self-Introspective Decoding: Alleviating Hallucinations for Large Vision-Language Models

**Fushuo Huo**[1][*]**, Wenchao Xu**[2][†]**, Zhong Zhang**[3]**, Haozhao Wang**[4]**, Zhicheng Chen**[5]**, Peilin Zhao**[3][†]

[1]Department of Computing, The Hong Kong Polytechnic University
[2]Division of Integrative Systems and Design, Hong Kong University of Science and Technology
[3]Tencent AI Lab, [4]Huazhong University of Science and Technology, [5]Tsinghua University
`wenchaoxu.ru@gmail.com,masonzhao@tencent.com`

## Abstract

Hallucination remains a significant challenge in Large Vision-Language Models (LVLMs). To alleviate this issue, some methods, known as contrastive decoding, induce hallucinations by manually disturbing the raw vision or instruction inputs and then mitigate them by contrasting the outputs of the original and disturbed LVLMs. However, these holistic input disturbances sometimes induce potential noise and also double the inference cost. To tackle these issues, we propose a simple yet effective method named *Self-Introspective Decoding* (SID). Our empirical investigations reveal that pre-trained LVLMs can introspectively assess the importance of vision tokens based on preceding vision and text (both instruction and generated) tokens. Leveraging this insight, we develop the Context and Text-aware Token Selection (CT$^2$S) strategy, which preserves only the least important vision tokens after the early decoder layers, thereby adaptively amplify vision-and-text association hallucinations during auto-regressive decoding. This strategy ensures that multimodal knowledge absorbed in the early decoder layers induces multimodal contextual rather than aimless hallucinations, and significantly reduces computation burdens. Subsequently, the original token logits subtract the amplified fine-grained hallucinations, effectively alleviating hallucinations without compromising the LVLMs' general ability. Extensive experiments illustrate that SID generates less-hallucination and higher-quality texts across various metrics, without much additional computation cost.

## 1 Introduction

Recent advancements in Large Language Models (LLMs) (Touvron et al., 2023a; Bai et al., 2023a; Chiang & Li, 2023; Touvron et al., 2023b; Meta, 2024) have demonstrated great success over the past few years. Many efforts have been made to extend LLMs to Large Vision-Language Models (LVLMs) (Ye et al., 2023; Li et al., 2023a; Bai et al., 2023b; Li et al., 2023c; Dai & et al, 2023; Liu et al., 2024b; Bavishi et al., 2023; Young et al., 2024; Li et al., 2024), achieving impressive performance across various vision tasks (Li et al., 2023b; Zhang et al., 2023) as well as more complex tasks like content comprehension (Lai et al., 2024) and generation (Geng et al., 2024).

Despite their extraordinary versatility, LVLMs face a significant challenge known as the 'hallucination'. Concretely, hallucinated texts are fluent and semantically coherent but contain incorrect statements about the given image, e.g., generating irrelevant or meaningless responses, identifying inaccurate colors, numbers, and locations of objects not present in the image (Huang et al., 2024). This flaw in LVLMs poses a significant risk for real-world applications to become trustworthy AI assistants. For instance, in model-assisted computer-aided diagnosis scenarios (Wang et al., 2023), such misinterpretation of medical images could lead to serious medical accidents.

---

[*]This work is done when F. Huo works as an intern in Tencent AI Lab
[†]Corresponding author

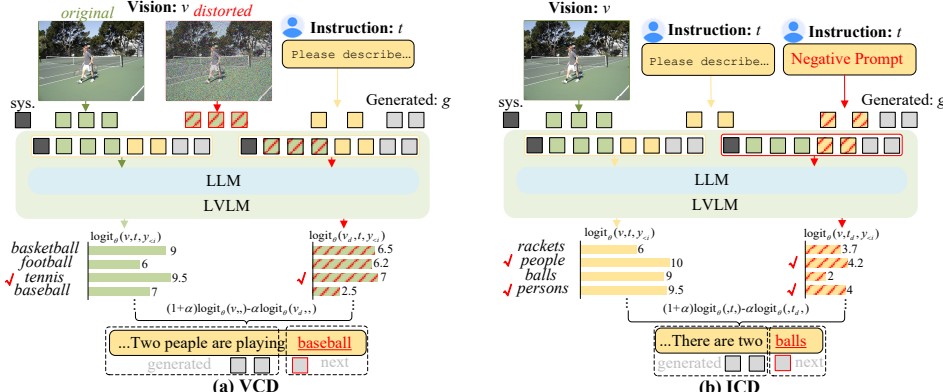

Figure 1: **Contrastive Decoding strategies:** (a) Visual Contrastive Decoding (VCD) (Leng et al., 2024) **manually** distort vision inputs. (b) Instruction Contrastive Decoding (ICD) (Wang et al., 2024a; Kim et al., 2024) also **manually** design noisy instruction (negative prompt). Detailed analyses are in Sec. 3.2. We ablate other modules like the vision encoder and tokenizer for clarity. $t$: 'Please describe this image in detail.'; sys.: system prompt. $g$: generated text tokens. $\alpha$ in Eq. 2 defaults to 1.

One mainstream approach to alleviating hallucinations in LVLMs involves developing training-free decoding strategies known as Contrastive Decoding (CD) (Leng et al., 2024; Favero et al., 2024; Wang et al., 2024a; Kim et al., 2024), which adjusts the next-token logits in a contrastive manner. Concretely, Vision CD (VCD) manipulates vision inputs with Gaussian noise (Leng et al., 2024) or directly ablates visual inputs (Favero et al., 2024) to amplify language priors. Instruction CD (ICD) (Wang et al., 2024a; Kim et al., 2024) designs negative prompt.[1] The rationale is that disturbed inputs significantly exacerbate hallucinations, and CD subtracts hallucinated concepts from the original distribution to mitigate hallucinations. *However*, input disturbances require elaborate designs for various downstream tasks, and the inference cost is inevitably doubled. *Moreover*, the contrastive distributions are *vision-and-text agnostic*, not necessarily amplify desired hallucinations but sometimes induce potential uncertainty noise for CD. Intuitive examples are illustrated in Fig. 1, and detailed analyses are in Sec. 3.2. In Fig. 1 (a) and (b), LVLMs directly infer the correct next token from multimodal inputs. For Vision CD, distorted vision input exacerbates hallucinated object logits such as *football* and *basketball*, while the holistic noise suppresses *baseball* to a low logit value. Consequently, VCD might compromise normal decoding. Similarly, for Instruction CD, LVLMs tend to refuse to answer negative prompts in open-end generation task (as seen in Fig. 5 and 10), and also suffer from potential uncertainty noise similar to VCD.

To address the aforementioned issues, we propose a novel decoding strategy called *Self-Introspective Decoding* (SID). Our empirical investigations reveal that pre-trained LVLMs can introspectively assess the importance of vision tokens adaptively, based on preceding vision and text (both instruction and generated) tokens. SID leverages this capability to amplify and then subtract *vision-and-text association* hallucinations by proposing token-level disturbances named Context and Text-aware Token Selection ($CT^2S$) strategy. This strategy induces multimodal contextual hallucinations, rather than aimless ones, by conducting token selection in the early decoder layers. In summary, our main contributions are three-fold:

• We re-think CD methods in LVLMs and attribute their failure cases to vision-and-text agnostic input distributions that induce potential uncertainty noise.

• To address this, we propose Self-Introspective Decoding (SID), where the $CT^2S$ strategy adaptively amplifies and then subtracts vision-and-text association hallucinations. This approach is grounded in our investigations that pre-trained LVLMs can introspectively assess visual importance informed by preceding tokens.

• Through comprehensive comparisons, we demonstrate that SID generates high-quality texts with fewer hallucinations. Additionally, SID significantly reduces inference cost of contrastive decoding.

## 2 RELATED WORK

We review recent advances in **Hallucination in Foundation Models**. More backgrounds about **Large Vision-Language Models** and **Decoding Strategy in LLMs** are in Appendix A.1.

---

[1]negative prompts like 'You are a confused object detector.' and 'Always respond with the opposite of what you're asked.' for different tasks.

**Hallucination in Foundation Models.** Hallucination, defined as the generation of irrelevant, factually incorrect, or meaningless text in a given context (Rohrbach et al., 2018; Zhang et al., 2024; Guan et al., 2024; Wu et al., 2024b), is a significant bottleneck in current foundation models. This issue can stem from overfitting specific patterns in the training data, a lack of understanding world knowledge, or an inability to effectively contextualize a given input (Ji et al., 2023). In the context of LLMs, hallucinations often manifest as generated content that conflicts with world knowledge or common sense. For LVLMs, the primary concern is whether the generated answer conflicts with the provided images. To mitigate the hallucination issue, several solutions have been proposed, including **robust instruction tuning with curated datasets** (Lee et al., 2022; Gunjal et al., 2024; Liu et al., 2024a; Zhao et al., 2024; Jiang et al., 2024; Yu et al., 2024b;a; Ma et al., 2024; Yue et al., 2024b), **post-hoc utilizing auxiliary analysis networks** (Manakul et al., 2023; Zhou et al., 2024; Yin et al., 2023; Chen et al., 2024b; Wu et al., 2024a; Feng et al., 2024), and **various decoding strategies** (Li et al., 2022; Chuang et al., 2024; Liu et al., 2024c; Leng et al., 2024; Favero et al., 2024; Wang et al., 2024a; Kim et al., 2024; Zhu et al., 2024). However, robust instruction tuning requires massive high-quality datasets and advanced GPU clusters, making it resource-intensive; Post-hoc utilizing auxiliary networks heavily rely on the auxiliary network, leading to high inference costs. As for decoding strategies, representative LVLMs hallucination alleviation methods (Leng et al., 2024; Favero et al., 2024; Wang et al., 2024a) manually disturb raw inputs to induce hallucinations then contrast them to alleviate the issue. However, holistic disturbing raw inputs might bring additional noise during contrastive decoding, and double the inference cost. In this paper, we propose an efficient Self-Introspective Decoding (SID) that induces and then mitigates vision-and-text association hallucination by token-level disturbances, greatly reducing the inference cost.

## 3 PRELIMINARY AND MOTIVATION

In the following, we first illustrate the generation paradigm of LVLMs to facilitate the understanding of SID. We then re-think the contrastive decoding in LVLMs and propose our motivation for SID.

### 3.1 PARADIGM OF LVLMs GENERATION

**Vision and Language Inputs.** The inputs to LVLMs consist of both image ($v$) and text ($t$). Generally, the raw images are commonly fed to the visual encoder, and then the cross-model projection module maps vision information into LLMs' input space, denoted as vision tokens $v = \{v_1, v_2...v_n\}$ ($n$ is the length of vision tokens). Similarly, text is processed by tokenizer and embedding modules, which is denoted as text tokens $t = \{t_1, t_2...t_m\}$ ($m$ is length of text tokens). Then, the image ($v$) and text ($t$) tokens are concatenated as the final input of LLMs.

**LVLMs Forward.** The backbone networks of LVLMs are pre-trained LLMs like Vicuna (Chiang & Li, 2023) and LLaMA 2 (Touvron et al., 2023b), parameterized by $\theta$. Given multimodal tokens $\{v, t\}$, LVLMs predict the next token probability ($y_i$) at $i$ time step in an auto-regressive manner following the methodology of LLMs, over the vocabulary set $\nu$:

$$p(y_i|v, t, y_{<i}) = \text{softmax}(logit_\theta(y_i|v, t, y_{<i})), y_i \in \nu \qquad (1)$$

**Next Token Decoding.** After obtaining the next token probability $p(y_i|v, t, y_{<i})$, different decoding strategies (Appendix A.1) are proposed to predict next token. The decoded token is concatenated to the last original input token, for the next round of generation until the end of the generation process.

### 3.2 RE-THINKING CONTRASTIVE DECODING IN LVLMs

Following the seminal works (Li et al., 2022) in natural language processing, which introduced the Contrastive Decoding (CD) mechanism to enhance coherence and informativeness by considering the differences between expert and amateur models, various studies have adapted this strategy to LVLMs by distorting the visual or instruction inputs for contrastive purposes. As the vision and instruction contrastive processes are symmetrical, we use visual contrastive decoding as an example. The contrastive decoded probability of next-token ($p_{cd}$) can be generally formulated as follows:

$$p_{cd}(y_i|v, v_d, t, y_{<i}) = \text{softmax}[(1+\alpha)logit_\theta(y_i|v, t, y_{<i}) - \alpha logit_\theta(y_i|v_d, t, y_{<i})] \qquad (2)$$

where $d$ and $\alpha$ indicate distortion operation and hyperparameter, respectively. *Generally*, CD methods employ an adaptive plausibility constraint to calibrate the entire output distribution, preventing

Table 1: **Efficacy Analyses on CD strategies** on MSCOCO dataset. The *Random* setting means objects absent from the image are chosen randomly, while the *Adversarial* setting prioritizes co-occurring objects which are not present in the image. Results are from (Leng et al., 2024) or the average of three running times conducted on LLaVA-1.5 7B for fair comparisons. $\alpha$ in Eq. 2 defaults to 1 and Eq. 3 has no effect on the greedy decoding.

| Setting | Method | Greedy | | Sampling | |
|---|---|---|---|---|---|
| | | Accuracy ↑ | F1 Score ↑ | Accuracy ↑ | F1 Score ↑ |
| *Random* | Normal | $88.8_{\pm0.05}$ | $88.6_{\pm0.08}$ | $84.9_{\pm0.03}$ | $83.2_{\pm0.01}$ |
| | VCD | $87.8_{\pm0.02}$ | $87.9_{\pm0.06}$ | $87.73$ | $83.28$ |
| | w/o Eq. 3 | - | - | $83.3_{\pm0.04}$ | $82.2_{\pm0.02}$ |
| | ICD | $87.9_{\pm0.04}$ | $88.1_{\pm0.02}$ | $86.9_{\pm0.03}$ | $85.2_{\pm0.04}$ |
| | w/o Eq. 3 | - | - | $82.7_{\pm0.02}$ | $81.8_{\pm0.03}$ |
| | **Ours** | $\mathbf{89.3}_{\pm0.08}$ | $\mathbf{89.5}_{\pm0.02}$ | $\mathbf{88.8}_{\pm0.03}$ | $\mathbf{88.7}_{\pm0.02}$ |
| | w/o Eq. 3 | - | - | $87.2_{\pm0.01}$ | $88.0_{\pm0.02}$ |
| *Adversarial* | Normal | $79.3_{\pm0.05}$ | $80.9_{\pm0.09}$ | $78.7_{\pm0.03}$ | $78.9_{\pm0.02}$ |
| | VCD | $80.9_{\pm0.06}$ | $81.0_{\pm0.04}$ | $80.88$ | $81.33$ |
| | w/o Eq. 3 | - | - | $76.2_{\pm0.04}$ | $76.0_{\pm0.04}$ |
| | ICD | $80.2_{\pm0.03}$ | $81.3_{\pm0.01}$ | $79.1_{\pm0.02}$ | $80.4_{\pm0.04}$ |
| | w/o Eq. 3 | - | - | $75.4_{\pm0.02}$ | $76.4_{\pm0.04}$ |
| | **Ours** | $\mathbf{83.3}_{\pm0.07}$ | $\mathbf{82.5}_{\pm0.06}$ | $\mathbf{82.6}_{\pm0.05}$ | $\mathbf{82.1}_{\pm0.06}$ |
| | w/o Eq. 3 | - | - | $82.2_{\pm0.03}$ | $81.9_{\pm0.01}$ |

implausible outputs from the augmented distribution (Li et al., 2022; Chuang et al., 2024; Leng et al., 2024; Favero et al., 2024; Wang et al., 2024a; Kim et al., 2024; Zhu et al., 2024):

$$\nu_{token}(y_{<i}) = \left\{ y_i \in \nu : p_\theta(y_i|v, t, y_{<i}) \geq \beta \max_\omega p_\theta(\omega|v, t, y_{<i}) \right\},$$
$$p_{cd}(y_i|v, v_d, t, y_{<i})) = 0, \text{ if } y_i \notin \nu_{token}(y_{<i})$$

(3)

where $\nu$ and $\nu_{token}$ are the output vocabulary and selected tokens. $\beta$ controls the strength of truncation, with larger $\beta$ indicating more aggressive truncation that retains only high-probability tokens.

However, we argue that manually disturbing raw inputs might not trigger the desired hallucinations, while holistic disturbances will bring uncertainty noise that compromises the normal decoding. To validate our claim, we analyze the performances of normal decoding, VCD, and ICD using the POPE (Li et al., 2023d) metric, under both **sampling** and **greedy** decoding settings. POPE quantitatively converts the hallucination evaluation into a binary classification problem by using the question format to prompt the model: `Is there a <object> in the image?`, with expected answers being `Yes` or `No`. From Table 1, under the **greedy** decoding setting, CD methods improve performance in the *adversarial* setting, which are more challenging as they prioritize co-occurring confusing objects. CD methods achieve this by exacerbating and subtracting hallucinated concepts from the original distribution. However, in *random* settings, where objects absent from the image are chosen randomly and are easily recognized, CD methods slightly underperform normal greedy decoding, which indicates that the correct token logit is somewhat compromised during contrastive decoding. In the **sampling** decoding setting, CD methods clearly outperform the normal sampling decoding. However, CD methods rely on the adaptive plausibility constraint (Eq. 3) to filter out low-probability tokens. Without Eq. 3, CD methods are inferior to normal decoding in both *random* and *adversarial* settings, validating that vision-and-text agnostic input distributions induces potential uncertainty noise after Eq. 2 (More validations on other benchmarks are in Table 11.). To address these issues, we propose a decoding strategy named *Self-Introspective Decoding* (SID). SID *adaptively* amplifies *vision-and-text association* hallucinations informed by generated tokens to guide LVLMs in exploring factuality. Details are illustrated in the Sec. 4 and Fig. 2.

## 4 METHODOLOGY

### 4.1 UNDERSTANDING THE SELF-INTROSPECTIVE PRE-TRAINED LVLMS.

LLMs (Bai et al., 2023a; Chiang & Li, 2023; Touvron et al., 2023b; Meta, 2024) have been scaled up to billions of paramters and pre-trained on trillions of tokens, endowing LLMs with encyclopedic ability like in-context learning (Wang et al., 2024b), zero (Kojima et al., 2022)/few-shot (Brown et al., 2020) ability. LVLMs extend LLMs to multimodal understanding capabilities by visual instruction tuning. Some works (Cao et al., 2023; Shang et al., 2024; Chen et al., 2024a) pointed out that vision information is redundant in LVLMs, and develop vision token reduction technologies

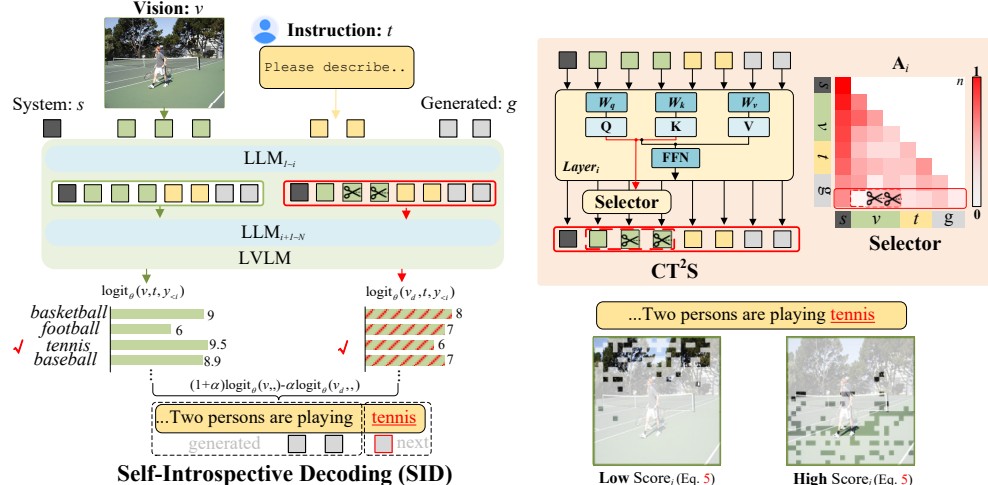

Figure 2: **Overview of Self-Introspective Decoding (SID).** CT$^2$S: Context and Text-aware Token Selection strategy. LLaVA-1.5 7B is utilized as an example to visualize visual tokens with low and high scores (Eq. 5).

to prune (Rao et al., 2021) and merge (Bolya et al., 2023) tokens guided by importance metrics without further re-training. Regarding the hallucination issue, we argue that vision tokens with low attention scores induce **vision-and-text association hallucination**. Formally, for the transformer block (Vaswani et al., 2017) in the auto-regressive decoder [2], vision ($v$), text instruction ($t$), and generated tokens ($g$) are concatenated and projected into three distinct vectors: the query vector $\mathbf{Q}$, the key vector $\mathbf{K}$, and the value vector $\mathbf{V}$, utilizing three linear transformations $W_q$, $W_k$, and $W_v$. The self-attention ($SA$) mechanism computes the relevance of each item to other items as follows:

$$\mathbf{R} = SA(\mathbf{Q}, \mathbf{K}, \mathbf{V}) = \mathbf{A} \cdot \mathbf{V},$$
$$\mathbf{A} = \text{softmax}(\frac{\mathbf{Q} \cdot \mathbf{K}^T}{\sqrt{d_l}} + M), \tag{4}$$

where $d_l$ represents the dimension of $\mathbf{Q}, \mathbf{K}, \mathbf{V}$, $M$ represents the casual mask. $\mathbf{A} \in R^{(b,h,n,n)}$, where $b$, $h$, and $n$ denote batch size, number of key-value heads, and total token number, respectively. We denote the $\mathbf{A}_i$ as the attention matrix after *Layer $i$* of LVLMs. We then calculate vision token importance scores ($\text{Score}_i(v)$) as shown in Fig. 2 (**Selector**) based on $\mathbf{A}_i$:

$$\text{Score}_i(v) = \frac{1}{h} \sum_{j=1}^{h} \mathbf{A}_i^{(\cdot, j, \cdot, \cdot)}[-1], \tag{5}$$

where $v$ means vision token indexes. Contrary to token pruning/merging (Rao et al., 2021; Bolya et al., 2023), we preserve a certain number of the least important vision tokens based on Eq. 5.

**Analyses.** Fig. 3 and 4 preliminarily validate the efficacy of $\text{Score}_i(v)$ qualitatively. In Fig. 3, the preserved **least** important tokens mainly reflect areas opposite to the query. For instance, when querying 'cup' in Fig. 3 (left), LVLMs focus on 'cup' in the foreground, thus preserving background tokens with low $\text{Score}_i(v)$. Conversely, LVLMs pay attention to background items when querying 'couch'. When querying existing items in Fig. 3 (right), vision tokens of unrelated regions are mainly preserved. For open-end generative tasks in Fig. 4, auto-regressive decoded tokens are generated based on preceding vision ($v$), instruction ($t$), and generated text ($g$) tokens. The preserved vision tokens are **adaptively adjusted** according to preceding tokens at each decoding step, primarily focusing on spurious related regions. More quantitative analyses are provided in Appendix A.7, where Table 12 illustrates that vision tokens with high $\text{Score}_i(v)$ greatly maintain original ability, while tokens with low $\text{Score}_i(v)$ reach nearly $50\%$ accuracy in binary classification. Above evaluations suggest that Eq. 5 effectively assesses the importance of vision tokens.

We further demonstrate the *open-end generated hallucinations* induced by ours, Vision Disturbance (VD) (Leng et al., 2024), and Instruction Disturbance (ID) (Wang et al., 2024a) in Fig. 5 and 10. The hallucinations we amplified are more vision-and-text association compared to VD, while LVLMs

---

[2]Here we illustrate the transformer block without KV Cache for better understanding.

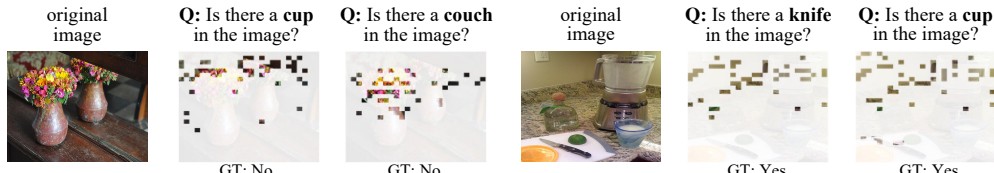

Figure 3: **Visualization Results** of the **least** important vision tokens on discrimination tasks informed by preceding vision and text tokens. LLaVA-1.5 7B with Layer $i = 3$ is utilized.

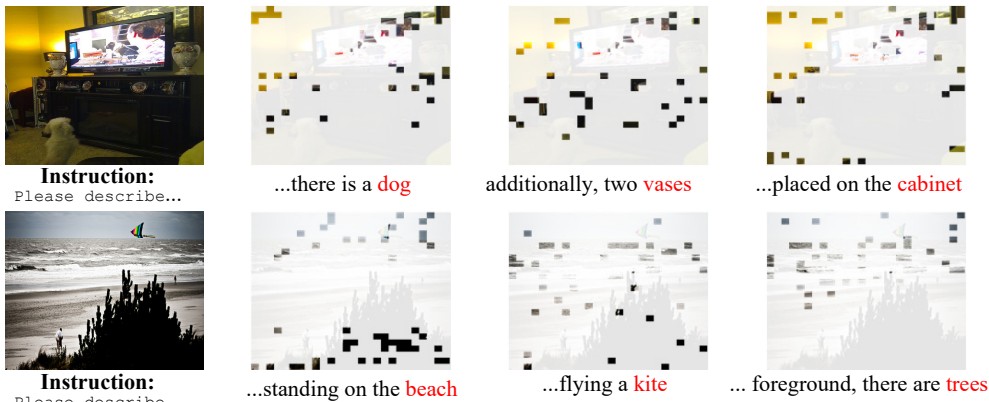

Figure 4: **Visualization Results of** *Adaptively* **Selecting** the **least** important vision tokens on open-end generative tasks informed by preceding vision and text tokens. LLaVA-1.5 7B with Layer $i = 3$ is utilized.

usually refuse to response to ID. Additionally, we demonstrate the quantitative results for discrimination and generation tasks with *VD and ID as inputs* in Appendix Table 12. Interestingly, VD and ID do not degrades much especially in discrimination tasks. Experiments imply that disturbed target logits still have the highest probability in most cases, and therefore, contrastive decoded target logits are not enhanced much after Eq. 2, while CD methods are susceptible to potential uncertainty noise.

## 4.2 CONTEXT AND TEXT-AWARE TOKEN SELECTION (CT²S) STRATEGY.

Based on the above investigations, we argue that to induce context- and text-aware hallucinations for contrastive decoding, only a small percentage of vision tokens with low attention scores should be preserved after the early decoder layers. To validate our claims, we conduct the following experiments: 1) In Vision Encoder (VE), we preserve tokens with low attention values between the [**CLS**] token and vision tokens in the penultimate layer, calculated as: $\mathbf{A} = \mathrm{softmax}(\frac{[\mathbf{CLS}] \cdot \mathbf{K^T}}{\sqrt{d_k}})$. 2) In the LLM decoder, we preserve tokens with low importance score (Eq. 5) across varying layers

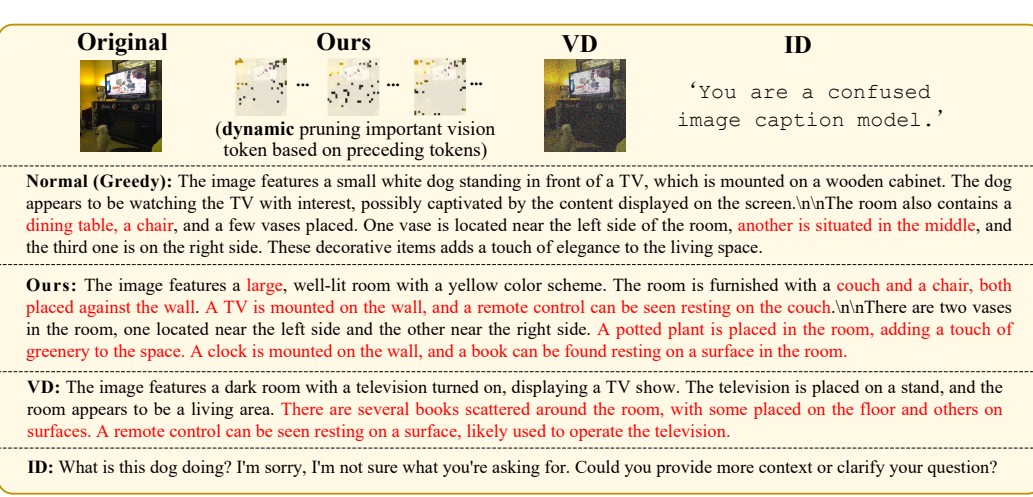

Figure 5: **Instance Illustration of Different Disturbance Results.** Examples are from MSCOCO inferred by LLaVA-1.5 7B with $i = 3$ and Top-k=50. Hallucinations are marked in red.

($i$). Additionally, we adjust the number of preserved vision tokens. As shown in Fig. 6, **firstly**, pruning vision tokens in VE based on [**CLS**] may not always yield positive gains, as the [**CLS**] token lacks information about instructions and generated texts, which are crucial for multimodal understanding. Specifically, pruning all vision tokens resembles VIG (Favero et al., 2024), which contrastively amplifies the vision importance over the language prior by ablating vision inputs. **Secondly**, aggressive pruning of vision tokens (i.e., 0%) after $Layer_{i=1}$ is not optimal. As the ideal induced hallucination distributions are *target-co-occurring* but suppress *target logits*, the loss of visual information for subsequent decoding results in visual context diminishing, which can lead to aimless hallucinations due to insufficient grounding in visual information. **Thirdly**, selecting tokens in the late decoder layers degrades contrastive decoding to normal decoding, as preceding layers of LVLMs already decode and understand multimodal information, which is consistent with LLMs'

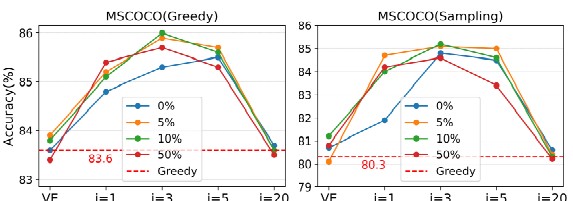

Figure 6: Analyses of varying $i$ and preserved ratios in CT²S. VE: vision encoder; $i$: $i$-th decoder layer.

early-exiting mechanisms (Schuster et al., 2022; Elhoushi et al., 2024). In summary, the proposed CT²S strategy selects Top-k least important vision tokens after the early layers based on attention score (Eq. 5), where the induced hallucinations are aware of both visual contexts and text information. Finally, following CD methods (Sec. 3.2), we contrastively subtract amplified vision-and-text association hallucinations for the next token prediction.

**Discussion.** Based on analyses of the self-introspective pre-trained LVLMs, could we enhance vision information informed by the proceeding vision and text tokens rather than utilizing the contrastive decoding? To explore this, we rewrite Eq. 2 as follows:

$$p_{add}(y_i|v, v \uparrow, t, y_{<i}) = \text{softmax}[(1-\alpha)logit_\theta(y_i|v, t, y_{<i}) + \alpha logit_\theta(y_i|v \uparrow, t, y_{<i})] \quad (6)$$

where we preserve vision tokens with high importance scores (Eq. 5) denoted as $v \uparrow$. From Table 2, we observe that enhancing vision information (i.e., Add) alleviates hallucinations to some extent, which also implicitly validates the efficacy of Eq. 5. However, in the adversarial setting, enhancing vision information does not bring much benefits compared to ours. Because our amplified hallucinations effectively associate co-occurring objects, reflected in **high logit values of hallucination token**, and then contrastively suppress them. In contrast, enhancing vision information primarily boosts the original prediction's **target** logits grounded in attention scores, which does not significantly improve discrimination, especially in the adversarial setting.

Table 2: **Analyses of Contrastive Decoding Mechanisms** on the POPE metric. Hyperparameters are consistent with CD settings.

| Setting | Random | Adversarial |
|---|---|---|
| Sampling | 84.7 | 78.7 |
| VCD | 87.7 | 80.9 |
| Add | **88.9** | 79.4 |
| **Ours** | 88.8 | **82.6** |
| Greedy | 88.8 | 79.1 |
| VCD | 87.8 | 80.9 |
| Add | 89.1 | 80.1 |
| **Ours** | **89.3** | **83.3** |

## 5 EXPERIMENTS

### 5.1 EXPERIMENTAL SETTINGS

**Models and Baselines.** We utilize four representative LVLMs: InstructBLIP (Dai & et al, 2023), Shikra (Chen et al., 2023a), LLaVA-1.5 (Liu et al., 2024b) at the 7B scale, and LLaVA-NeXT (Li et al., 2024) at the 8B scale. For detailed model descriptions and **results at larger scales**, please refer to Appendix A.2 and Table 14 in Appendix A.7. Since our method aims to propose *training-free* LVLM decoding strategies *without* the aid of auxiliary networks, we compare six decoding methods: Sampling (Top-p=1), Greedy, Dola (Chuang et al., 2024), and LVLM decoding strategies (VCD (Leng et al., 2024), ICD (Wang et al., 2024a), and OPERA (Huang et al., 2024)). For comprehensive comparisons, we apply VCD and ICD in both sampling (Top-p=1) and greedy decoding settings. Additionally, we validate SID with different decoding strategies, as detailed in Appendix A.7.

**Implementation Details.** As analyzed in Sec. 4.2, we set Layer $i$=3 and preserve top 10% least important vision tokens for Shikra, LLaVA-1.5, and LLaVA-NeXT and $i$=5 and top 10% least important vision tokens for Q-former based LVLMs (InstructBLIP) to induce fine-grained hallucinations. Hyperparameters in Eq. 2 and 3 follow VCD and ICD. More details are in Appendix A.2.

## 5.2 EVALUATION RESULTS

In this section, we follow previous methods (Leng et al., 2024; Wang et al., 2024a; Huang et al., 2024) to evaluate the SID on **CHAIR** (Rohrbach et al., 2018) and **POPE** (Li et al., 2023d) metrics. Besides manually designed metrics, we also leverage **GPT-4 assisted benchmark** (Zhao et al., 2024) to evaluate attribute, location, and relation hallucinations. **MME** (Fu et al., 2023) and **MM-Bench** (Liu et al., 2023b) benchmarks are employed to assess the LVLM's general ability. Moreover, **GPT4-V assisted evaluation** (Appendix A.4) on both hallucination alleviation and generated text quality, **more general benchmarks evaluation** (Appendix A.5) on MathVista (Lu et al., 2024), MMVet (Yu et al., 2024c), LLaVA-Bench (Liu et al., 2023a), MMMU (Yue et al., 2024a), and **case studies** (Appendix A.6) on LLaVA-Bench-in-the-Wild are in Appendix.

**CHAIR and POPE Evaluations.** CHAIR (Rohrbach et al., 2018) and POPE (Li et al., 2023d) are quantitative metrics to assess objection hallucinations of VLMs. Please refer to the Appendix A.3 for detailed descriptions of CHAIR and POPE. As for **CHAIR**, Following (Wang et al., 2024a; Huang et al., 2024; Yue et al., 2024b), we randomly select 500 images from the validation set of

Table 3: **Results on the CHAIR metric.** $^*$ and $^\star$ denote adopting the same sampling and greedy decoding strategies, respectively.

| Setting | LLaVA-1.5 | | InstructBLIP | | Shikra | | LLaVA-NeXT | |
|---|---|---|---|---|---|---|---|---|
| | $C_S\downarrow$ | $C_I\downarrow$ | $C_S\downarrow$ | $C_I\downarrow$ | $C_S\downarrow$ | $C_I\downarrow$ | $C_S\downarrow$ | $C_I\downarrow$ |
| Sampling | 51.3 | 16.8 | 51.0 | 24.2 | 48.9 | 14.7 | 42.6 | 14.1 |
| ICD$^*$ | 48.7 | 13.9 | 48.3 | 16.7 | 47.8 | 14.5 | 42.7 | 13.6 |
| VCD$^*$ | 48.0 | 14.3 | 47.9 | 17.2 | 48.1 | 13.8 | 41.3 | 12.9 |
| **Ours$^*$** | 45.0 | **11.7** | 43.6 | 13.1 | 46.0 | 12.9 | 38.4 | 11.4 |
| Greedy | 49.6 | 14.4 | 54.6 | 13.6 | 47.1 | 13.9 | 42.9 | 13.2 |
| Dola$^\star$ | 47.1 | 13.8 | 52.7 | 14.0 | 46.8 | 14.2 | 40.9 | 13.1 |
| OPERA | 45.2 | 12.7 | 47.4 | 12.9 | **44.4** | 13.6 | 39.4 | 11.8 |
| ICD$^\star$ | 47.4 | 13.9 | 46.3 | 15.3 | 47.3 | 14.1 | 42.1 | 12.6 |
| VCD$^\star$ | 46.8 | 13.2 | 44.0 | 13.6 | 47.8 | 14.0 | 41.1 | 12.9 |
| **Ours$^\star$** | **44.2** | 12.2 | **42.3** | **12.4** | 44.8 | **12.8** | **38.1** | **11.3** |

the MSCOCO (Lin et al., 2014) dataset and query different LVLMs with the prompt: `'Please describe this image in detail.'`. We set the max new tokens to 512 to generate responses for fair comparisons. As shown in Table 3, our method outperforms other baselines in most cases, validating the effectiveness of SID in open-end generation tasks. Compared to CD methods, SID *online adaptively* prunes attention-important vision tokens informed by instruction and generated text to induce fine-grained hallucinations for contrastive decoding during open-end text generations. For the **POPE** metric, which comprises three datasets, we average the results in Table 4. Our method performs best overall in *random*, *popular*, and *adversarial* sampling settings. Specifically, in the sampling decoding setting, SID surpasses the normal sampling decoding by a large margin in a train-free manner. SID also clearly outperforms CD methods (Dola, ICD, and VCD) because the self-introspective decoding strategy amplifies *vision-and-text association* hallucinations then subtracts them, rather than coarsely disturbing raw inputs. Additionally, owing to the context and text-aware token selection strategy, SID is more computation-efficient than CD methods, as analyzed in Table 6. Note that beam-search based OPERA (Huang et al., 2024) shows almost no gain in the POPE metric, primarily because answering the binary classification only requires a few tokens and selecting the best beam score in a decoded sequence ($N$=5) brings little improvement.

**GPT-4 Assisted Benchmark.** While CHAIR and POPE evaluate object-existence-level hallucinations, these metrics are unable to identify other types of hallucination, such as *positional*, *relational*, and *attribute* hallucinations. Therefore, the GPT-4 assisted benchmark (Zhao et al., 2024) utilizes the fine-grained object-level descriptions in the Visual Genome (VG) dataset (Krishna et al., 2017) as ground-truth and relies on the advanced GPT-4 to judge the fine-grained hallucinations and calculate Sentence-level Hallucination Ratio (SHR). Please refer to the Appendix A.3 for detailed implementations. Moreover, we employ n-gram fluency (n = 1 and 2) metrics to measure the smoothness of generated text, and the number of generated words/sentences per image (WPI/SPI) to compare the detailedness of generated texts. As shown in Fig. 7, SID achieves the best results in the SHR metric among the four LVLMs, outperforming others by a clear margin. Regarding the quality of the generated texts, Sampling decoding outperforms ours slightly in terms of 1-gram fluency and WPI. However, compared to other baselines, our approach alleviates hallucinations with minimal sacrifice in text generation quality regarding smoothness and detailness. For instance, OPERA generates text with fewer words and sentences due to penalization of the over-trust mechanism, and VCD impairs text fluency, possibly arising from the holistic and fixed disturbance of contrastive inputs.

**MME and MMBench Evaluations.** Besides, we test on two popular LVLMs' general ability benchmarks: MME and MMBench. MME comprises ten subtasks to evaluate models' perceptual

Table 4: **Average results on the POPE metric.** $^*$ and $^\star$ denote adopting the same sampling and greedy decoding strategies, respectively. Results are from the papers or re-implemented based on official codes.

| Setting | | *Random* | | *Popular* | | *Adversarial* | |
|---|---|---|---|---|---|---|---|
| Model | Decoding | Accuracy↑ | F1 Score↑ | Accuracy↑ | F1 Score↑ | Accuracy↑ | F1 Score↑ |
| LLaVA-1.5 | Sampling | 84.77 | 82.28 | 79.98 | 79.34 | 76.03 | 76.26 |
| | ICD$^*$ | 87.51 | 83.28 | 83.15 | 83.91 | 79.13 | 80.41 |
| | VCD$^*$ | 86.84 | 86.83 | 82.65 | 83.37 | 77.31 | 79.28 |
| | **Ours**$^*$ | 88.91 | 88.84 | 83.97 | 85.42 | 82.54 | 81.98 |
| | Greedy | 88.81 | 88.52 | 82.76 | 83.36 | 79.11 | 80.92 |
| | Dola$^\star$ | 87.94 | 87.97 | 83.87 | 84.68 | 80.35 | 81.21 |
| | OPERA | 88.85 | 88.67 | 82.77 | 83.40 | 79.16 | 80.93 |
| | ICD$^\star$ | 87.97 | 87.84 | 84.03 | 84.22 | 80.21 | 80.97 |
| | VCD$^\star$ | 87.02 | 86.96 | 83.53 | 84.56 | 78.12 | 80.16 |
| | **Ours**$^\star$ | 89.46 | 89.62 | 85.13 | 85.94 | 83.24 | 82.21 |
| InstructBLIP | Sampling | 80.42 | 80.94 | 76.09 | 77.65 | 72.37 | 75.42 |
| | ICD$^*$ | 85.78 | 85.73 | 81.12 | 82.25 | 76.82 | 78.99 |
| | VCD$^*$ | 84.11 | 84.13 | 79.96 | 80.80 | 76.32 | 78.08 |
| | **Ours**$^*$ | 86.56 | 85.94 | 80.26 | 81.75 | 77.64 | 80.41 |
| | Greedy | 84.56 | 83.75 | 78.23 | 79.16 | 74.58 | 76.34 |
| | Dola$^\star$ | 84.67 | 83.38 | 78.21 | 79.19 | 75.69 | 77.98 |
| | OPERA | 84.57 | 83.74 | 78.24 | 79.15 | 74.59 | 76.33 |
| | ICD$^\star$ | 84.36 | 83.82 | 77.88 | 78.70 | 75.17 | 77.23 |
| | VCD$^\star$ | 84.52 | 83.63 | 78.04 | 78.45 | 75.95 | 77.76 |
| | **Ours**$^\star$ | 87.23 | 86.90 | 81.16 | 82.57 | 78.51 | 81.26 |
| Shikra | Sampling | 81.42 | 82.46 | 79.60 | 80.78 | 73.85 | 76.39 |
| | ICD$^*$ | 82.34 | 82.82 | 78.17 | 80.43 | 74.96 | 77.68 |
| | VCD$^*$ | 82.31 | 82.73 | 79.34 | 80.93 | 75.61 | 77.96 |
| | **Ours**$^*$ | 83.87 | 83.94 | 80.26 | 82.07 | 77.85 | 78.94 |
| | Greedy | 83.00 | 83.19 | 81.39 | 81.90 | 76.69 | 78.31 |
| | Dola$^\star$ | 82.87 | 82.98 | **82.42** | 82.50 | 76.85 | 78.09 |
| | OPERA | 83.05 | 83.20 | 81.40 | 81.89 | 76.73 | 78.31 |
| | ICD$^\star$ | 82.67 | 82.64 | 80.73 | 81.58 | 75.98 | 78.43 |
| | VCD$^\star$ | 82.96 | 82.63 | 80.68 | 81.27 | 76.94 | 78.32 |
| | **Ours**$^\star$ | 84.46 | 84.62 | 82.38 | **82.73** | 78.67 | **79.34** |
| LLaVA-NeXT | Sampling | 86.32 | 83.11 | 82.27 | 81.03 | 77.32 | 76.96 |
| | ICD$^*$ | 87.32 | 84.03 | 83.62 | 83.54 | 80.31 | 80.41 |
| | VCD$^*$ | 86.97 | 86.71 | 83.07 | 83.65 | 79.42 | 80.28 |
| | **Ours**$^*$ | 89.16 | 88.92 | 84.38 | 85.76 | 82.95 | 81.98 |
| | Greedy | 89.37 | 88.82 | 83.68 | 84.62 | 80.08 | 80.74 |
| | Dola$^\star$ | 88.73 | 88.67 | 84.56 | 84.96 | 80.32 | 80.68 |
| | OPERA | 89.36 | 88.80 | 83.65 | 84.60 | 80.10 | 80.75 |
| | ICD$^\star$ | 87.40 | 87.96 | 84.11 | 83.79 | 80.94 | 80.67 |
| | VCD$^\star$ | 87.83 | 87.09 | 82.68 | 83.55 | 79.61 | 81.20 |
| | **Ours**$^\star$ | 90.05 | 89.97 | 86.13 | 85.69 | 84.06 | 82.95 |

Figure 7: **GPT-4 assisted benchmark** (Zhao et al., 2024). **Hallucination** (SHR), **fluency** (1&2-gram), and **detailness** (WPI and SPI) aspects are compared. Larger areas mean better performances. VCD and ours adopt the same sampling decoding. Please zoom in for details.

capabilities and four subtasks for assessing recognitive abilities in the form of the yes/no question. MMBench systematically evaluates twenty ability dimensions of LVLMs. We present the results of LLaVA-1.5 7B as a representative in Table 5, SID can maintain and improve the multimodal ability on LVLMs benchmarks. In contrast, other CD methods tend to compromise the general multimodal ability. More general benchmark evaluations are in Appendix A.5.

Table 5: **LVLM benchmark evaluations.** DoLa, ICD, VCD, and SID employ the same greedy decoding.

| | Greedy | Sampling | DoLa | ICD | VCD | OPERA | **SID** |
|---|---|---|---|---|---|---|---|
| MME | $1510.8_{\pm1.2}$ | $1471.5_{\pm5.6}$ | $1480.7_{\pm1.3}$ | $1473.2_{\pm1.2}$ | $1488.5_{\pm0.8}$ | $1515.2_{\pm1.1}$ | $\mathbf{1520.4_{\pm0.9}}$ |
| MMbench | $64.4_{\pm.22}$ | $63.9_{\pm.81}$ | $63.7_{\pm.22}$ | $63.0_{\pm.24}$ | $63.8_{\pm.22}$ | $64.4_{\pm.13}$ | $\mathbf{65.0_{\pm.23}}$ |

## 6 ABLATION ANALYSES

In this section, we conduct ablation analyses on the computation efficiency and hyperparameter sensitivity. More analyses about **Token Selection Strategies Analyses**, **Larger-scale Backbone**, **Other Decoding Strategies**, and **Visual Enhancing Decoding Strategy** are in Appendix A.7.

| Methods | Time ↓ | Memory↓ | Accuracy↑ |
|---|---|---|---|
| Normal | 494 | 15673 | 79.11 |
| VCD | 904 | 16753 | 78.12 |
| ICD | 974 | 16843 | 80.21 |
| OPERA | 2643 | 21943 | 79.16 |
| **Ours**$_{40\%}$ | 704 | 15809 | 83.11 |
| **Ours**$_{10\%}$ | 668 | 15767 | 83.24 |

Table 6: **Efficiency Comparisons** on NVIDIA V100. $_{10\%}$ and $_{40\%}$ mean tokens preserved ratios.

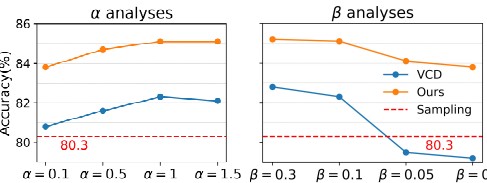

Figure 8: **Hyperparameter Sensitivity of** $\alpha$ **and** $\beta$ with POPE metric (under the sampling decoding).

**Computation Efficiency.** One primary concern of hallucination alleviation decoding methods is the computational burden. We evaluate the whole dataset inference time (seconds) and peak GPU memory (MB) on the LLaVA-1.5 7B under the POPE adversarial setting, as shown in Table 6. Contrastive Decoding (CD) methods (Leng et al., 2024; Wang et al., 2024a) involve constructing distorted raw inputs, resulting in **twice** the inference complexity. OPERA (Huang et al., 2024) is based on beam-search decoding and maintains a set of beams to enlarge the candidate range. Additionally, roll back mechanism in the retrospection-reallocation strategy further exacerbates computational complexity. Our SID induces vision-and-text association hallucinations by pruning *large-ratio* attention-important tokens in the *early layers*, which greatly reduces the inference time of CD up to ∼30%.

**Hyperparameter Sensitivity.** Beyond the sensitivity analyses in Fig. 6, we validate the robustness of SID concerning $\alpha$ and $\beta$ of Eq. 2 and 3, compared to the contrastive decoding methods (i.e., VCD) on LLaVA-1.5 7B. From Fig. 8 (left), it is evident that as $\alpha$ decreases, the contrastive decoding mechanism diminishes. However, SID still achieves pleasant results, while VCD degrades close to Sampling when $\alpha$=0.1, as the CT$^2$S strategy induces informative *vision-and-text association* hallucinations. When $\alpha$ increases, VCD degrades to some extent because holistic input disturbance does not always trigger contextual-related hallucination and might exacerbate uncertainty noise. Regarding $\beta$, a larger $\beta$ indicates more aggressive truncation of the output vocabulary. Fig. 8 (right) shows that VCD's performance heavily relies on large $\beta$ to retain only high-probability tokens. With mild or no adaptive plausibility constraint (Eq. 3), VCD performs worse than the sample decoding strategy due to output logits influenced by distorted visual inputs. Ours is robust to the $\beta$ setting as the CT$^2$S strategy induces discriminative contrastive logits to generate plausible tokens.

## 7 CONCLUSION AND FUTURE WORK

In this paper, we firstly re-think contrastive decoding in LVLMs and empirically find that vision-and-text-agnostic input disturbances in CD do not always amplify desired hallucinations rather than induce potential uncertainty noise. To mitigate these issues, we propose a training-free decoding strategy named Self-Introspective Decoding (SID). By developing Context and Text-aware Token Selection (CT$^2$S) strategy, SID amplifies *vision-and-text association* hallucinations to guide LVLMs in contrastive decoding, thereby improving faithfulness. Extensive experiments validate the effectiveness and robustness of SID.

**Future Work: 1)** As the pruning ratios and layer are set manually, we consider training the external network to automatically determine optimal hyperparameters, inspired by (Chen et al., 2023b). In addition, to enhance the interpretability of hallucination alleviations, we consider resorting to pretrained analysis networks to intuitively locate spurious related vision regions. **2)** Moreover, given that SID amplifies fine-grained hallucinations, we consider leveraging the CT$^2$S strategy to automatically construct high-quality negative instruction for robust visual instruction tuning rather than relying on expensive GPT-4 (Liu et al., 2024a; Zhao et al., 2024). Note that the self-generated hallucination dataset ensures *style consistency*, which is crucial for preference learning (Zhao et al., 2024). Codes are available at `https://github.com/huofushuo/SID`.

## 8    ACKNOWLEDGEMENTS

This work is supported by two grants from the Research Grants Council of the Hong Kong Special Administrative Region, China (Project No. PolyU15222621, PolyU15225023). We would like to thank Tencent AI Lab for supporting Fushuo Huo as a student researcher during his internship.

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

# A  APPENDIX

**Overview.** More related work are described in Appendix A.1, including backgrounds about **Large Vision-Language Models** and **Decoding Strategy in LLMs**. The detailed experimental settings are in Appendix A.2 including **Model Details** and **Implementation Details**. Evaluation metric details, including **CHAIR Evaluations**, **POPE Evaluations**, and **GPT-4 Assisted Evaluations** are introduced in Appendix A.3. We demonstrate the **GPT4-V Assisted Evaluation** in Appendix A.4 for comprehensive evaluations. Moreover, we illustrate the results on **More Benchmarks** including MathVista (Lu et al., 2024), MMVet (Yu et al., 2024c), LLaVA-Bench (Liu et al., 2023a), MMMU (Yue et al., 2024a) to evaluate the complex reasoning ability in Appendix A.5. We visualize some **Case Study** from the LLaVA-Bench-in-the-Wild dataset for qualitative analysis in Appendix A.6. More ablation experiments and analyses including **More Analyses on Other Baselines**, **More Analyses on Adaptive Plausibility Constraint (Eq. 3)**, **Quantitative Attention Scores Analyses**, **Token Selection Strategies Analyses**, **Larger-scale LVLM Backbone**, **Adopting Other Decoding Strategies**, and **Visual Enhancing Decoding Strategy** are illustrated in Appendix A.7

## A.1  MORE BACKGROUNDS

**Large Vision-Language Models.** Motivated by the success of Large Language Models (LLMs) (Touvron et al., 2023a; Bai et al., 2023a; Chiang & Li, 2023; Taori et al., 2023; Touvron et al., 2023b; Meta, 2024), recent studies have extended LLMs to multimodal regions and provided Large Vision-Language Models (LVLMs) (Liu et al., 2023a; Zhu et al., 2023; Chen et al., 2023a; Ye et al., 2023; Li et al., 2023a; Bai et al., 2023b; Li et al., 2023c; Dai & et al, 2023; Liu et al., 2024b; Bavishi et al., 2023; Chen et al., 2024c; Li et al., 2024) powered by pre-trained LLMs. LVLMs understand and generate diverse content in a more comprehensive way by integrating user instruction and vision inputs. LLaVA (Liu et al., 2023a) connects open-set vision encoder with LLMs (i.e., Vicuna (Chiang & Li, 2023)) by instruction tuning with elaborated language-image instruction-following data. Then, LLaVA-1.5 (Liu et al., 2024b) develops the vision-language connector that is data-efficient and powerful for better multimodal understanding. Shikra (Chen et al., 2023a) further incorporates grounding data and trains the model to understand the grounding knowledge in the given images. BLIP-2, InstructBLIP, and MiniGPT-4 (Li et al., 2023c; Dai & et al, 2023; Zhu et al., 2023) introduce a learnable querying transformer to fusion multimodal features and largely reduce image tokens. Fuyu (Bavishi et al., 2023) proposes a vanilla decoder-only architecture without the vision encoder and adapter that makes it easier to understand, scale, and deploy. InternVL (Chen et al., 2024c) proposes three simple but effective improvements, including a strong vision encoder, dynamic high-resolution, and high-quality bilingual dataset. Recently, built on SOTA open-source LLaMA 3 (Meta, 2024) and increasing the input vision resolution to $4\times$ more pixels, LLaVA-NeXT (Li et al., 2024) exhibits excellent multimodal capabilities. Despite the impressive results, all of the above LVLMs suffer from serious trustworthy issues (Liu et al., 2024f;e), especially the hallucination problem, and we mainly conduct experiments on advanced LVLMs, including InstructBLIP, Shikra, LLaVA-1.5, and LLaVA-NeXT.

**Decoding Strategy in LLMs.** Selecting decoding strategies in language models is crucial, as it determines how models generate text. Greedy decoding selects the highest probability next token at each step but might lead to less varied text. Beam search (Graves, 2012) is an accumulated-score-based decoding strategy. It maintains a set of beams to enlarge the candidate range and finally selects the best one in beams, which is more sophisticated than greedy decoding. Sampling decoding generates the next words by randomly selecting from the output distribution, where Top-k sampling (Fan et al., 2018) samples from Top-k likely tokens (Fan et al., 2018) and brings diversity but sometimes induces less coherent outputs. Top-p (Nucleus) sampling (Holtzman et al., 2020) improves Top-k sampling that considers the dynamic number of words that reach the probability p, achieving a balance between randomness and relevance. As demonstrated in Tables 1 and 3, while greedy decoding consistently surpasses sampling decoding in terms of hallucination metrics, most open-source and closed-source LVLMs default to using sampling decoding to promote diverse and coherent chat interactions. Therefore, it is practically meaningful to consider the constraints in Eq. 3 when analyzing sampling decoding. Recently, to alleviate the hallucination issue, DoLa (Chuang et al., 2024) decoding emphasizes the knowledge of mature layers and downplays that of pre-mature layers. OPERA (Huang et al., 2024) is established on beam-search decoding strategy and finds the interesting phenomenon of high-probability co-occurrence between the hallucination

and the knowledge aggregation patterns. OPERA penalizes 'Over-Trust Logit' in the beam score to alleviate aggregation patterns. Self-Introspective Decoding (SID) can be seamlessly integrated into different decoding strategies to mitigate hallucinations without sacrificing text generation quality, such as diversity, coherence, and relevance.

## A.2 DETAILED EXPERIMENTAL SETTINGS

**Models Details.** To validate the effectiveness of our SID, we conduct experiments on four representative LVLMs: InstructBLIP (Dai & et al, 2023), Shikra (Chen et al., 2023a), LLaVA-1.5 (Liu et al., 2024b), and LLaVA-NeXT (Li et al., 2024). InstructBLIP employs Q-former (Li et al., 2023c) to condense image tokens to 32, as a result, we are unable to visualize the dynamic token pruning process of InstructBLIP like Fig. 3 and 4. Shikra, LLaVA-1.5, and LLaVA-NeXT directly leverage linear projection layers as vision-language connectors to align multimodal features. Shikra and LLaVA-1.5 encode 256 and 576 image tokens to LVLMs. LLaVA-NeXT increases the input vision resolution by $4\times$ to capture more visual details, resulting in $4\times$ more encoded vision tokens than LLaVA-1.5. All LVLMs utilize pre-trained vision encoders like CLIP (Radford et al., 2021) vision encoder, as well as pre-trained LLMs as language decoders, such as Vicuna v1.1 (Chiang & Li, 2023), LLaMA 2 (Touvron et al., 2023b), and recently released LLaMA 3 (Meta, 2024). We provide results at the 7 Billion (B) scale, and larger-scale results are in the Appendix A.7.

**Implementation Details.** For sampling and greedy decoding, we adopt the default hyperparameter settings. As for Dola (Chuang et al., 2024), it is designed to alleviate hallucinations (i.e., improve factuality) of LLM by contrasting the differences in logits obtained from projecting the later layers versus premature layers. Dola is sensitive to the premature layer selection, we adapt Dola to LVLMs, following OPERA (Huang et al., 2024) to utilize "0,2,4,6,8,10,12,14" as the indexes of candidate premature layers and "32" as the index of the mature layer. The repetition penalty is set to 1.2, as Dola suggests. OPERA, VCD, and ICD are proposed for LVLMs and we adopt the default settings. For fair comparisons, SID's hyperparameters of Eq. 2 and 3 follow VCD and ICD. Moreover, we apply SID, VCD, and ICD in both sampling (Top-p=1) and greedy decoding settings for comprehensive comparisons. Note that due to amplified fine-grained hallucinations, SID is more robust to hyperparameters compared to other CD methods (Sec. 5). Experiments are performed on NVIDIA V100/A100 GPUs.

## A.3 EVALUATION METRIC DETAILS

**CHAIR Evaluations.** The Caption Hallucination Assessment with Image Relevance (CHAIR) (Rohrbach et al., 2018) metric is specially designed to assess objection hallucinations in the image caption tasks. Concretely, CHAIR quantifies the degree of hallucinations in a generated image caption by calculating the proportion of all objects mentioned in the caption that are not present in the ground truth label pool. There are two common variants of CHAIR: CHAIRi ($C_I$) and CHAIRs ($C_S$), which evaluate the degree of object hallucination in the instance and sentence level, respectively. These two metrics are formulated as follows:

$$C_I = \frac{|\{\text{hallucinated objects}\}|}{|\{\text{all mentioned objects}\}|}, \ C_S = \frac{|\{\text{captions with hallucinated objects}\}|}{|\{\text{all captions}\}|} \quad (7)$$

The smaller the value of $C_I$ and $C_S$, the better the hallucination alleviation performance.

**POPE Evaluations.** The Polling-based Object Probing Evaluation (POPE) (Li et al., 2023d) was recently developed to assess hallucination problems in LVLMs. POPE queries the LVLMs with the template: `Is there a <object> in the image?` The ratio between queries about existing and no-existing objects is balanced (i.e., 50%-50%). This benchmark consists of three sampling settings: *random*, *popular*, and *adversarial*, each differing in the construction of negative samples. Specially, in the *random* setting, objects that are not present in the image are selected at random. The *popular* setting selects missing objects from the high-frequency pool, whereas in the *adversarial* setting, co-occurring objects that are not present in the image are prioritized. POPE consists of three different datasets, including MSCOCO (Lin et al., 2014), A-OKVQA (Schwenk et al., 2022), and GQA (Hudson & Manning, 2019). POPE involves 500 images from each dataset with six questions each, ultimately yielding 27,000 query-answer pairs. Accuracy and F1 score are chosen as evaluation metrics. The larger the value of Accuracy and F1 score, the better the hallucination alleviation performance.

Table 7: **GPT4-V assisted hallucination evaluations** (Huang et al., 2024; Yin et al., 2023). VCD and ours adopt the same sampling decoding strategy. $C$: correctness; $D$: detailedness

| Setting | LLaVA-1.5 | | InstructBLIP | | Shikra | | LLaVA-NeXT | |
|---|---|---|---|---|---|---|---|---|
| | $C\uparrow$ | $D\uparrow$ | $C\uparrow$ | $D\uparrow$ | $C\uparrow$ | $D\uparrow$ | $C\uparrow$ | $D\uparrow$ |
| Sampling | 5.18 | 5.79 | 4.73 | 5.10 | 5.03 | **5.17** | 5.34 | 5.67 |
| **Ours** | **5.97** | **6.01** | **5.62** | **5.16** | **5.78** | 5.10 | **6.47** | **5.85** |
| VCD | 5.46 | 5.63 | 4.98 | 5.21 | 5.31 | 5.24 | 5.92 | 5.47 |
| **Ours** | **6.16** | **5.9**4 | **5.37** | **5.46** | **5.61** | **5.29** | **6.12** | **5.78** |
| OPERA | **6.16** | 5.57 | 5.29 | 4.86 | 5.34 | 4.87 | 6.11 | 5.24 |
| **Ours** | 6.15 | **5.94** | **5.76** | **5.42** | **5.97** | **5.88** | **6.63** | **6.23** |

**GPT-4 Assisted Evaluations.** Besides object-existence-level hallucinations evaluated by CHAIR and POPE, GPT-4 assisted benchmark (Zhao et al., 2024) utilizes the fine-grained object-level description in the Visual Genome (VG) dataset (Krishna et al., 2017) as ground-truth and relies on the advanced GPT-4 to judge the detailed (such as *positional*, *relational*, and *attribute*) hallucinations and calculate Sentence-level Hallucination Ratio (SHR). With the generated sentences and manually annotated factual information, GPT-4 is prompted to evaluate whether existing hallucinations sentence by sentence. The prompt template is provided in Appendix Fig. 11. Following (Zhao et al., 2024), we utilize 200 images from the VG dataset and set max new tokens to 512, with the prompt of 'Please describe this image in detail.' We conduct experiments on sampling decoding strategies and representative LVLMs decoding strategies: VCD (Leng et al., 2024) and OPERA (Huang et al., 2024).

### A.4 GPT4-V Assisted Evaluation.

To further analyze the hallucinations and text quality for open-end generation tasks, following (Huang et al., 2024; Yin et al., 2023), we utilize the strong multi-modal assistant GPT4-V, which simultaneously processes input from vision and text modalities. We strictly follow (Huang et al., 2024), which utilizes 500 images from the MSCOCO dataset and prompts LVLM: 'Please describe this image in detail.' with the maximum number of 512. To mitigate the impact of the sequential order fed to GPT4-V, we simultaneously compare the generated texts obtained from two decoding methods and instruct GPT4-V to judge the correctness and detailedness score on a scale of 0-10 based on the input image. The detailed GPT4-V prompt is in Appendix Fig. 12. We set up three representative pairs of comparison experiments: greedy decoding and ours, CD-based VCD (Leng et al., 2024) and ours, and OPERA (Huang et al., 2024) and ours. As shown in Table 7, our SID achieves the best results in terms of most metrics. Concretely, our method improves correctness by about 15-20% compared to sampling decoding while not compromising the detailedness level. Compared to advanced hallucination mitigation methods VCD and OPERA, SID generates text with obvious more details and better mitigates the hallucination issue. Since the perceptual and reasoning capabilities of GPT4-V are very close to those of humans, the results of the GPT4-V evaluation reflect, to some extent, the strong performance of the compared methods in terms of mitigating hallucinations and generating text quality from a human perceptual perspective.

### A.5 More general benchmarks evaluation.

To further validate the general ability of self-introspective decoding strategy, we conduct experiments on complex reasoning benchmarks, including MathVista (Lu et al., 2024), MMVet (Yu et al., 2024c), LLaVA-Bench (Liu et al., 2023a), MMMU (Yue et al., 2024a). **MathVista** is a robust mathematical reasoning evaluation benchmark that includes a series of challenging tasks requiring detailed deep visual recognition and compositional reasoning skills. MathVista is composed of 6,141 examples derived from 31 multimodal datasets involving mathematics. We utilize the 'test-mini' subset for testing. **MMVet** is a benchmark designed to evaluate large multimodal models on complex multimodal tasks. It outlines six fundamental vision-language capabilities: Recognition, Knowledge, OCR, Spatial Awareness, Language Generation, and Math, and assesses their integration across 16 emerging tasks. MMVet utilizes an LLM-based evaluator for open-ended outputs, providing consistent scoring across diverse question types and answer styles. This benchmark offers insights into the abilities of various large multimodal model paradigms and models, extending beyond mere performance ranking. **LLaVA-Bench** is a rigorous evaluation suite aimed at testing the visual-language alignment and instruction-following capabilities of LVLMs. Here, we utilize the LLaVA-Bench (In-the-Wild) subset, which includes 24 varied images from different domains, such

Table 8: **LVLM General Benchmark Evaluations.** $^\star$ denotes employing greedy decoding strategy.

|            | Greedy | VCD$^\star$ | OPERA | SID$^\star$ |
|------------|--------|-------------|-------|-------------|
| MathVista  | 27.1   | 26.3        | 27.1  | **27.4**    |
| MMVet      | 31.1   | 30.2        | 31.1  | **31.2**    |
| LLaVA-Bench| 63.4   | 63.6        | 64.3  | **68.7**    |
| MMMU       | 32.6   | 31.0        | 32.6  | **33.4**    |

Table 9: **Object-level and Attribute-level Hallucination Evaluations.** $^*$ and $^\star$ denote adopting the same sampling and greedy decoding strategies. Experiments are conducted on LLaVA-1.5 7B.

|            | Object-level ↑ | | Attribute-level ↑ | | Total |
|------------|-----------|-------|----------|-------|-------|
|            | Existence | Count | Position | Color |       |
| Sampling   | 171.7     | 120.8 | 112.6    | 151.0 | 556.1 |
| Dola$^*$   | 173.2     | 122.4 | 115.2    | 152.8 | 563.6 |
| VCD$^*$    | 174.4     | 124.1 | 119.4    | 153.6 | 571.5 |
| Ours$^*$   | 180.5     | 130.7 | 123.8    | 154.5 | 589.5 |
| Greedy     | 182.3     | 130.3 | 126.8    | 155.7 | 594.1 |
| Dola$^\star$ | 180.1   | 127.4 | 119.3    | 154.4 | 581.2 |
| VCD$^\star$ | 179.5    | 128.1 | 123.8    | 155.5 | 586.9 |
| Ours$^\star$ | 183.9   | 132.2 | 127.8    | 155.9 | 599.8 |

as memes, paintings, and sketches, with a total of 60 questions. **MMMU** serves as a benchmark for assessing multimodal models on tasks that span multiple disciplines, necessitating college-level subject knowledge, and deliberate reasoning. It includes 11,500 questions across 30 subjects and 183 sub-fields, focusing on advanced perception and reasoning in domain-specific knowledge areas. We compare SID with two representative decoding methods (i.e., VCD and OPERA) on LLaVA-1.5 7B. Table 8 indicates that our SID enhances reason abilities, particularly in the LLaVA-Bench benchmark. However, VCD slightly degrades LVLM's complex reasoning ability, as reflected in MathVista, MMVet, and MMMU benchmarks. OPERA has little gains in discrimination tasks. These results are consistent with Table 5 (MME and MMbench benchmarks), Table 3, and Table 4. We also illustrate experimental results of Existence, Count, Position, and Color subsets of the MME benchmark in Table 9, which evaluate the object-level and attribute-level hallucinations. Without the assistance of auxiliary analysis networks, our method achieves +33.4 and +18.0 compared to sampling and VCD on MME subsets, validating the effectiveness in alleviating object- and attribute-level hallucinations.

## A.6 CASE STUDY.

In addition to using crafted metrics (CHAIR and POPE), GPT-4/GPT4-V-aided evaluations, and MME (Fu et al., 2023) and MMBench (Liu et al., 2023b) benchmarks, we qualitatively present several case studies of SID's hallucination alleviation ability from LLaVA-Bench-in-the-Wild dataset (Liu et al., 2023a). As illustrated in Appendix Fig. 13, 14, and 15, SID effectively mitigates hallucination in these challenging scenes by dynamically amplifying vision-and-text association hallucinations. Meanwhile, it preserves the detailness of each image. As we propose a training-free decoding method that does not rely on auxiliary analysis networks, it inherently carries over the existing weaknesses of LVLMs. Intuitive case studies, as illustrated in Appendix Fig. 13, 14, and 15, reveal that SID still generates some hallucinations, particularly in finer details such as eye color and vehicle identification specifics. These failures may be attributed to the vision encoder's relatively limited visual perception ability. For future work, it is promising to integrate SID with InternVL (Chen et al., 2024c), which scales the vision encoder up to 6B, or consider leveraging auxiliary analysis networks like Grounding DINO (Liu et al., 2024d) or OWLv2 (Minderer et al., 2023) to mitigate LVLMs' internal weaknesses.

## A.7 MORE ABLATION ANALYSES

**More Analyses on Other Baselines.**

Table 10: **Comparisons with tuning-based and auxiliary networks-based baselines.** All methods adopt the greedy decoding strategy. LURE and HA-DPO adopt MiniGPT-4 13B and LLaVA-1.5 7B, respectively.

| | POPE↑ | CHAIRs↓ | CHAIRi↓ | MME↑ |
|---|---|---|---|---|
| Greedy | 78.6 | 55.4 | 14.2 | 805.7 |
| LURE | 78.7 | 55.1 | 14.0 | 846.2 |
| +VCD | 78.3 | 55.0 | 14.3 | 813.4 |
| +SID | 82.5 | 52.1 | 13.6 | 854.9 |
| SID | 82.2 | 51.9 | 13.2 | 838.1 |
| Greedy | 83.6 | 49.6 | 14.4 | 1510.8 |
| HA-DPO | 85.4 | 44.7 | 13.6 | 1522.3 |
| +VCD | 83.0 | 46.1 | 13.8 | 1500.6 |
| +SID | 86.2 | 43.8 | 12.4 | 1532.7 |
| SID | 85.9 | 44.2 | 12.2 | 1525.3 |

As SID is a **training-free** plug-and-play decoding strategy that do not requires **robust instruction tuning** with **curated datasets** (Lee et al., 2022; Gunjal et al., 2024; Liu et al., 2024a; Zhao et al., 2024; Jiang et al., 2024; Yu et al., 2024b;a; Ma et al., 2024; Yue et al., 2024b) and **post-hoc** utilizing **auxiliary analysis networks** (Manakul et al., 2023; Zhou et al., 2024; Yin et al., 2023; Chen et al., 2024b; Wu et al., 2024a; Feng et al., 2024), we do not compare with above two types of methods for fairness. To further validate the robustness of SID, we compare SID with the auxiliary analysis networks-based method (i.e., LURE (Zhou et al., 2024)) and the robust instruction tuning-based method (i.e., HA-DPO (Zhao et al., 2024)) and apply the plug-and-play SID to LURE and HA-DPO on POPE, CHAIR, and MME benchmarks, following their official implementations as shown in Table 10. Experimental results indicate that SID outperforms LURE and HA-DPO in most cases. Additionally, seamlessly integrating SID with LURE and LRV can significantly enhance performance.

**More Analyses on Adaptive Plausibility Constraint.**

To further analyze the effectiveness of Eq. 3, we test contrastive decoding-based methods with and without adaptive plausibility constraint (Eq. 3) on open-end generation task (CHAIR (Rohrbach et al., 2018)) and more comprehensive MME benchmark (Fu et al., 2023). As shown in Table 11, the adaptive plausibility constraint (Eq. 3) greatly affects generated texts, and most metrics underperform vanilla sampling decoding without the adaptive plausibility constraint (Eq. 3). The proposed SID avoids potential uncertainty noise by inducing fine-grained hallucinations, greatly maintaining the performances when ablating Eq. 3.

Table 11: **Efficacy Analyses of Eq. 3** on LLaVA-1.5 7B. All methods adopt the sampling decoding strategy, and results are the average of three running times with three random seeds.

| | CHAIRs↓ | CHAIRi↓ | MME↑ |
|---|---|---|---|
| Sampling | 51.3 | 16.8 | 1471.5 |
| ICD | 48.7 | 13.9 | 1479.7 |
| w/o Eq. 3 | 52.2 | 16.6 | 1448.5 |
| VCD | 48.0 | 14.3 | 1481.9 |
| w/o Eq. 3 | 51.9 | 17.2 | 1457.1 |
| SID | 45.0 | 11.7 | 1487.3 |
| w/o Eq. 3 | 46.5 | 12.2 | 1484.6 |

**Attention Scores and Token Selection Strategies Analyses.**

**Quantitative Attention Scores Analyses.** Here we quantitatively analyze vision attention score in terms of POPE (Li et al., 2023d) and CHAIR (Rohrbach et al., 2018) metrics. Concretely, we select the top-100 and least-100 important vision tokens out of a total of 576 vision tokens of LLaVA-1.5 7B based on attention score based on Eq. 5. Visual and Instruction Disturbance (VD and ID) are also employed as inputs for analyses. Quantitative results in Table 12 illustrate that 100 out of 576 vision tokens with high attention scores greatly maintain original ability, while low attention score tokens reach almost 50% accuracy for the binary classification problem, which indicates attention scores are a good indicator for vision token importance. As for VD and ID, disturbance in raw input does not obviously harm the LVLMs' discrimination ability, as indicated by the POPE metric. However, VD and ID significantly compromise the open-end generation tasks reflected by the CHAIR metric (LVLMs tend to refuse to ID as shown in Fig. 5 and 10).

**Token Selection Strategies Analyses.** To validate the effectiveness of SID in selecting low attention scores to induce vision-and-text association hallucination, we further conduct quantitative

Table 12: **Efficacy Analyses on Vision Token Attention Scores** with POPE metric on MSCOCO dataset and CHAIR metric. We select the Top-100 and Least-100 important vision tokens out of a total of 576 vision tokens of LLaVA-1.5 7B, based on Eq. 5 ($i$=3). **VD**: Visual Disturbance; **ID**: Instruction Disturbance.

| Setting | Random | | Adversarial | | CHAIRs ↓ | CHAIRi ↓ |
|---|---|---|---|---|---|---|
| | Accuracy ↑ | F1 Score ↑ | Accuracy ↑ | F1 Score ↑ | | |
| **Greedy** | 88.8 | 88.6 | 79.3 | 80.9 | 49.6 | 14.4 |
| +Top-100 | 85.6 | 83.9 | 77.1 | 76.3 | 52.7 | 15.2 |
| +Least-100 | 55.3 | 66.1 | 54.0 | 65.3 | 63.2 | 38.7 |
| **+VD** | 88.0 | 87.6 | 78.9 | 79.8 | 56.7 | 16.9 |
| **+ID** | 88.2 | 87.7 | 79.1 | 80.1 | - | - |

Table 13: **Analyses of Different Token Selection Strategies** with POPE on MSCOCO dataset and CHAIR metrics. We select the high importance scores (Eq. 5) of vision tokens (-**Top**) and random vision tokens (-**Random**) for contrastive decoding. Experiments are conducted on LLaVA-1.5 7B.

| Setting | Random | | Adversarial | | CHAIRs ↓ | CHAIRi ↓ |
|---|---|---|---|---|---|---|
| | Accuracy ↑ | F1 Score ↑ | Accuracy ↑ | F1 Score ↑ | | |
| **Greedy** | 88.8 | 88.6 | 79.3 | 80.9 | 49.6 | 14.4 |
| Ours | 89.3 | 89.5 | 83.3 | 82.5 | 44.2 | 12.2 |
| -High | 87.0 | 87.3 | 76.5 | 79.4 | 57.9 | 25.6 |
| -Random | 88.4 | 87.2 | 80.9 | 81.5 | 48.6 | 13.5 |
| AVISC | 88.4 | 88.1 | 79.8 | 80.5 | 45.3 | 14.7 |
| **Sampling** | 84.9 | 83.2 | 78.7 | 78.9 | 51.3 | 16.8 |
| Ours | 88.8 | 88.7 | 82.6 | 82.1 | 45.0 | 11.7 |
| AVISC | 87.9 | 87.9 | 77.5 | 79.6 | 46.6 | 12.5 |

experiments under different vision token selection strategies with the same preserved vision token number and Layer $i$=3 as ours. Table 13 shows that vision tokens with high attention scores degrade obviously, as it does not amplify contextual hallucinations rather than retain original important information. Contrastive decoding does not benefit from subtracting hallucinations amplified by the disturbed inputs rather than suffers from the potential disturbance noise. Selecting random vision tokens brings improvements in the *adversarial* setting because randomly selected vision tokens amplify the over-reliance on statistical bias and language priors, similar to Vision CD (Leng et al., 2024) and Instruction CD (Wang et al., 2024a). However, token-level random disturbance also induces uncertainty noise, resulting in the inferior performance in the *random* setting to greedy decoding. Moreover, AVISC (Woo et al., 2024), in contrast to ours, preserves outlier high attention tokens (named 'blind token') and substracts output logits to counteract the overemphasis of 'blind token.' In this way, AVISC promotes balanced consideration of all tokens to alleviate hallucinations. However, Table 13 illustrates that Top-100 vision tokens with high attention scores can largely maintain the original performance. 'blind token' tends to have a high probability of target class logits, and contrastive decoding does not improve the target class's probability while might bring extra noise. Table 13 indicates AVISC still degrades the greedy decoding to some extent, which indicates the attentional vision re-calibration strategy of AVISC induces some annoying noise. Overall, these experiments further validate the rationality of our token selection strategy based on attention sores.

Table 14: **Results on Larger-scale Backbones.** Sampling decoding is adopted and results average of three running times.

| Methods | POPE | | CHAIR | |
|---|---|---|---|---|
| | Accuracy | F1 Score | $C_S$ | $C_I$ |
| LLaVA-1.5 | 81.60 | 80.31 | 49.6 | 16.1 |
| +VCD | 82.67 | 81.46 | 46.7 | 16.4 |
| +OPERA | 82.32 | 81.10 | **43.3** | 13.6 |
| **+Ours** | **84.75** | **83.17** | 43.5 | **12.7** |
| InstructBLIP | 77.26 | 79.23 | 50.8 | 19.7 |
| +VCD | 79.77 | 80.27 | 47.9 | 17.6 |
| +OPERA | 80.31 | 80.91 | 42.5 | 14.3 |
| **+Ours** | **81.97** | **82.21** | **41.7** | **13.3** |

**Larger-scale LVLM Backbones.**

We validate the effectiveness of SID in terms of 13B scale backbones on LLaVA-1.5 and Instruct-BLIP architectures. We choose POPE (Li et al., 2023d) and CHAIR (Rohrbach et al., 2018) to

validate the hallucination issues in both discrimination and open-end generation tasks. Table 14 shows that SID remains effective as backbone networks scale up.

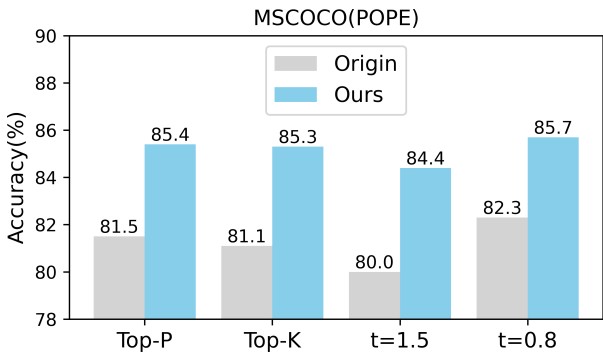

Figure 9: **Results of different decoding strategies.**

**Adopting Other Decoding Strategies.**

Meanwhile, besides direct sampling and greedy decoding, we conduct experiments on LLaVA-1.5 7B using the MSCOCO dataset with various decoding strategies, including Top-p sampling ($p$=0.9), Top-k sampling ($k$=50), Top-k sampling with varying temperature ($k$=50, $t$=1.5 and 0.8). Figure 9 shows that, regardless of the sampling strategy adopted, the application of SID consistently helps to alleviate hallucinations and improve the overall performance of LVLMs. This consistency highlights the versatility and effectiveness of SID across different sampling strategies.

**Visual Enhancing Decoding Strategy.**

Although LVLMs can accurately recognize visual elements, LVLMs have difficulty fully interpreting those elements in the context of the input cue and effectively linking that recognition to their internal knowledge. We follow Visual Description Grounded Decoding (VDGD) (Ghosh et al., 2024) by first generating a detailed description of the image and appending it as a prefix to the instruction. The prompt template is adopted from (Ghosh et al., 2024): `<image> I have been given this image to complete the task described as: inst. To help me complete the task, describe the given image in detail. In the case of real-world scenes, please include all foreground and background objects in the description, their properties (like color, shape, etc.), their relations with other objects, their count, and all other components in the image. In case of non-real-world scenes, like charts, graphs, tables, etc., please describe the table, mention all numbers (if any), mention the written text, and all other details.` Experiments are performed on hallucination evaluation benchmarks( i.e., POPE and CHAIR) and the general ability benchmark (i.e., MMbench). We re-implement VDGD based on official codes on LLaVA-1.5 7B. Table 15 demonstrates the effectiveness of VDGD (Ghosh et al., 2024) in LVLM's hallucination alleviation and general reasoning ability. However, the grounding visual descriptions, generated by LVLMs themself, enhance the visual perception reasoning capabilities while might inevitably contain hallucinations. Therefore, VDGD is inferior in the POPE (**adversial**) subset, which prioritizes **co-occurring** objects which are not present in the image. Meanwhile, VDGD shares somewhat similar motivations in enhancing vision information via Eq. 6 as we analyzed. The experiments in Table 2 are consistent with the above results, indicating that boosting the vision information is effective in mitigating hallucinations but is less effective in complex (i.e., adversarial) environments.

Table 15: **Comparisons with Visual Enhancing Strategy (VDGD).** $\star$ denotes employing greedy decoding strategy.

| | POPE↑ (random) | POPE↑ (adversarial) | CHAIRs↓ | CHAIRi↓ | MMbench↑ |
|---|---|---|---|---|---|
| Greedy | 88.8 | 79.1 | 49.6 | 14.4 | 64.4 |
| OPERA | 88.9 | 79.2 | 45.2 | 12.7 | 64.4 |
| VDGD$\star$ | 89.0 | 79.4 | 46.7 | 13.7 | **65.2** |
| SID$\star$ | **89.3** | **83.3** | **44.2** | **12.2** | 65.0 |

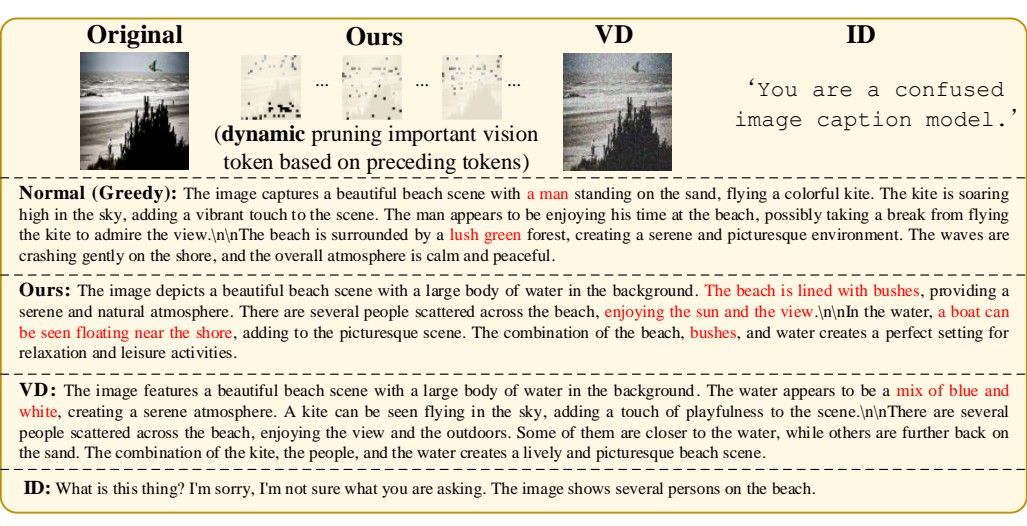

**Original**     **Ours**     **VD**     **ID**

... ... ... (**dynamic** pruning important vision token based on preceding tokens)

'You are a confused image caption model.'

**Normal (Greedy):** The image captures a beautiful beach scene with a man standing on the sand, flying a colorful kite. The kite is soaring high in the sky, adding a vibrant touch to the scene. The man appears to be enjoying his time at the beach, possibly taking a break from flying the kite to admire the view.\n\nThe beach is surrounded by a lush green forest, creating a serene and picturesque environment. The waves are crashing gently on the shore, and the overall atmosphere is calm and peaceful.

**Ours:** The image depicts a beautiful beach scene with a large body of water in the background. The beach is lined with bushes, providing a serene and natural atmosphere. There are several people scattered across the beach, enjoying the sun and the view.\n\nIn the water, a boat can be seen floating near the shore, adding to the picturesque scene. The combination of the beach, bushes, and water creates a perfect setting for relaxation and leisure activities.

**VD:** The image features a beautiful beach scene with a large body of water in the background. The water appears to be a mix of blue and white, creating a serene atmosphere. A kite can be seen flying in the sky, adding a touch of playfulness to the scene.\n\nThere are several people scattered across the beach, enjoying the view and the outdoors. Some of them are closer to the water, while others are further back on the sand. The combination of the kite, the people, and the water creates a lively and picturesque beach scene.

**ID:** What is this thing? I'm sorry, I'm not sure what you are asking. The image shows several persons on the beach.

Figure 10: **Instance Illustration** of different disturbance results. Examples are from MSCOCO (Lin et al., 2014) inferred by LLaVA-1.5 7B with $i = 3$ and Top-k=50. Hallucinations are marked in red.

---

**GPT-4 Prompt**

Please help me judge if the comment of this image is hallucination or correct.

I will give you a list of region description of a image. The format is [x1, y1, x2, y2]: region description, where [x1, y1, x2, y2] is the bounding box of the region. Highly overlapping bounding boxes may refer to the same object. This is the ground truth information of the image. Besides, I give you some factual information about the content of the image (which is 100% accurate). Your judgement should base on this information. However, this information only descibe the objects in the region of image, so it cannot descibe the subjective part of the image, e.g., atmosphere, style, emotion. In that case, you can return "Cannot judge".

Also, I will give you a list of comments of the image for you to judge if it is hallucination. Please give a judgement one by one along with the reason.

Your output should be:
Judgement:
1. hallucination or correct or cannot judge: <reason>
2. ...

Here are the region descriptions of the image:
{}

Factual Information:
{}

Here is the comment for you to judge (hallucination, correct, or cannot judge):
{}

Figure 11: **Prompts of GPT-4 for evaluations.**

> **GPT4-V Prompt**
> - - - - - - - - - - - - - - - - - - - - - - - - - - - - - - - - - - - - - - - - - - -
>
> You are required to score the performance of two AI assistants in describing a given image. You should pay extra attention to the hallucination, which refers to the part of descriptions that are inconsistent with the image content, such as claiming the existence of something not present in the image or describing incorrectly in terms of the counts, positions, or colors of objects in the image. Please rate the responses of the assistants on a scale of 1 to 10, where a higher score indicates better performance, according to the following criteria:
>
> 1: Accuracy: whether the response is accurate with respect to the image content. Responses with fewer hallucinations should be given higher
> scores.
> 2: Detailedness: whether the response is rich in necessary details. Note that hallucinated descriptions should not count necessary details.
> Please output the scores for each criterion, containing only two values indicating the scores for Assistant 1 and 2, respectively. The two scores
> are separated by a space. Following the scores, please provide an explanation of your evaluation, avoiding any potential bias and ensuring that
> the order in which the responses were presented does not affect your judgment.
> [Assistant 1]
> {}
> [End of Assistant 1]
> [Assistant 2]
> {}
> [End of Assistant 2]
> Output format:
> Accuracy: <Scores of the two answers>
> Reason:
> Detailedness: <Scores of the two answers>
> Reason:

Figure 12: **Prompts of GPT4-V for evaluations.**

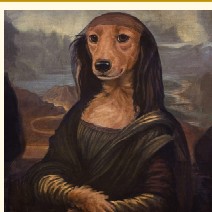

**Instruction**

(max new tokens=512)

**Please describe this image in detail.**

**Greedy:** The image features a dog wearing a gown, possibly painted as a mona lisa painting. The dog is looking at the camera with a curious expression, displaying its unique attire. The gown appears to be a burgundy color and complements the dog's outfit.\n\nThe setting of the painting includes a pond and a rock formation in the background, adding a sense of depth and atmosphere to the artwork. Additionally, there is another small figure, possibly a dog or a human, in the upper-left corner of the picture.

**OPERA:** The image features a painting of a dog wearing a Renaissance-style dress, with a hood over its head. The dog appears to be a poodle, and it is sitting in a chair, giving the impression of a human-like figure. The painting is quite detailed, capturing the dog's facial expression and the intricate design of the Renaissance dress. The dog's attire and the overall composition of the painting create a unique and interesting visual experience.

**VCD:** The image is a painting of a woman wearing a Renaissance-style dress and a hood, resembling a dog. The woman has a dog's face painted on her forehead, and she is wearing a hat. The painting captures the unique and whimsical blend of human and canine features.\n\nIn the background, there is a boat visible, adding a sense of depth and context to the scene. The woman's attire and the dog's face on her forehead create a playful and imaginative atmosphere.

**Ours:** The painting features a dog wearing Renaissance paint makeup and a veil, posing as a Mona Lisa-style figure. The dog's face is adorned with a mustache and hat, giving it a unique and artistic appearance. The dog's sitting position is also painted to resemble a human's, adding to the creative and humorous nature of the painting.\n\nIn the background, there is a river visible, which complements the overall composition of the painting. The dog's position and the artistic elements create a captivating and amusing scene.

Figure 13: **Case Study from LLaVA-Bench-in-the-Wild** on LLaVA-1.5 7B. Hallucinations are marked in red.

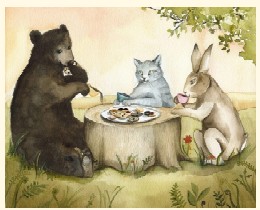

**Instruction**

(max new tokens=512)

**Please describe this image in detail.**

**Greedy:** The image features a group of three animals sitting around a dining table, enjoying a meal together. The animals include a large brown bear, a cat, and a rabbit. They are all seated on the grass, with the bear on the left side, the cat in the middle, and the rabbit on the right side of the table.\n\nThe table is set with various food items, including a cake, a bowl, and a cup. The cake is placed in the center of the table, while the bowl and cup are positioned closer to the edge. The scene appears to be a whimsical and playful depiction of the animals sharing a meal.

**OPERA:** The image depicts a whimsical scene of a bear, a cat, and a rabbit sitting around a dining table, enjoying a meal together. The bear is on the left side of the table, the cat is in the middle, and the rabbit is on the right side. Each animal has a plate of food in front of them.\n\nThe table is set with various utensils such as forks, knives, and spoons, as well as cups and bowls. There are also a couple of cakes placed on the table, adding to the delightful atmosphere of the gathering.\n\nIn the background, a potted plant can be seen, adding a touch of greenery to the scene.

**VCD:** The image features a whimsical scene of three animals sitting around a dining table, enjoying a meal together. A brown bear is sitting on one side of the table, while a cat is seated on the opposite side, and a rabbit is sitting on the far right side of the table.\n\nVarious food items are placed on the table, including a cup, a bowl, and several cakes. The cakes are spread across the table, with one near the center, another closer to the rabbit, and the third one near the cat. The table is set in a lush green field, giving the scene a peaceful and inviting atmosphere.

**Ours:** The watercolor illustration features three animals sitting around a dining table, enjoying cookies and doughnuts. A large brown bear is on one side of the table, while a gray cat and a yellowish rabbit are on the other side. The cat has blue eyes and is drinking from a blue cup, while the rabbit holds a pink cup as well.\n\nThe table is topped with a plate of various desserts, including multiple cookies and doughnuts. The scene is set in a field surrounded by green grass and trees, giving the whole watercolor a natural and whimsical atmosphere.

Figure 14: **Case Study from LLaVA-Bench-in-the-Wild** on LLaVA-1.5 7B. Hallucinations are marked in red.

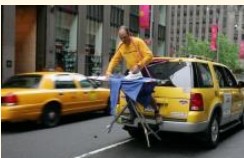

**Instruction**

(max new tokens=512)

**Please describe this image in detail.**

**Greedy:** The image features a man standing on the back of a yellow taxi cab, holding a blue shirt. He appears to be ironing the shirt while the taxi is in motion. The taxi is driving down a busy street, with other vehicles such as a truck and a car visible in the scene.\n\nThere are also a few potted plants scattered around the area, adding some greenery to the urban environment. Additionally, a traffic light can be seen in the background, indicating that the taxi is driving through an intersection.

**OPERA:** In the image, a man wearing a yellow shirt is standing on the back of a yellow taxi, ironing clothes. The taxi is driving down a busy city street, surrounded by other vehicles such as cars and trucks. One car is positioned behind the taxi on the left side, while another car can be seen further down the street on the right side. A truck is also visible on the right side of the street.\n\nThere are several potted plants placed along the street, adding a touch of greenery to the urban environment. One potted plant is located near the center of the scene, while two others are positioned closer to the right side of the street.

**VCD:** The image features a man ironing clothes on the back of a yellow SUV taxi. He is standing on a clothes iron, which placed on top of a metal frame mounted on the back of the vehicle. The man is focused on his task, and the scene is quite unusual as it takes place on a busy city street.\n\nThere are several other vehicles in the scene, including a truck and a car on the left side of the image, and another car on the right side. Additionally, there are two potted plants visible in the scene, one near the center and the other on the left side.

**Ours:** In the image, a man is ironing clothes while riding on the back of a yellow SUV taxi. He is standing on a portable desk, which is strapped to the back of the vehicle. The taxi is driving down a busy city street, surrounded by various vehicles, including a yellow taxi and a truck.\n\nThe man is wearing a yellow shirt and shorts, and he appears to be focused on his task. Green trees can be seen in the background, adding a touch of greenery to the busy urban environment. The man's unconventional choice of transportation adds a unique element to the bustling cityscape.

Figure 15: **Case Study from LLaVA-Bench-in-the-Wild** on LLaVA-1.5 7B. Hallucinations are marked in red.

