# OpenReview forum: "Self-Introspective Decoding: Alleviating Hallucinations for Large Vision-Language Models"
_ICLR.cc/2025/Conference — ICLR 2025 Poster_

### Official Review · Reviewer_yTiw · 2024-10-31

**Soundness:** 3
**Presentation:** 3
**Contribution:** 3
**Rating:** 6
**Confidence:** 5

**Summary:**

1) The main contribution os the paper is proposing that some vision tokens with low attention scores can induce hallucinations while decoding.
2) The paper backs this up by conducting experiments to prove that low score vision tokens focus mainly on unrelated regions in the image.
3) The paper does extensive ablations on till which layer to prune vision tokens and comparison with other baselines.

**Strengths:**

1) Presentation of the paper is clear and straightforward.
2) The paper does a good set of experiments to support their argument that low importance score tokens induce hallucinations.
3) The paper does sufficient ablations and evaluations to prove its approach.

**Weaknesses:**

1) An important baseline the paper has missed is a simple but effective one, how would the comparison look if we append the image description before asking a question? Ask the LVLM to describe the image and append it to the question and pass through the model again and see if it has any improvement. Some research on this includes - Visual Description Grounding Reduces Hallucinations and Boosts Reasoning in LVLMs (Ghosh et al, 2024).

2) A few more important benchmark comparisons are needed - MathVista, MMVet, LLaVA-Bench, MMMU

**Questions:**

Refer to weakness section.

---

> ### Author Response · Authors · 2024-11-21
> **Response to the Reviewer yTiw [Concerns 1]**
>
> Thank you for the thoughtful concerns.
> We are thrilled that the reviewer considers our paper to have a clear presentation, extensive experiments and ablations, and well-supported motivations.
> We appreciate the opportunity to add more discussions and experiments on more baselines and benchmarks.
>
> $\textbf{Concern 1:}$ **An important baseline (Ghosh et al [1]) needs to be compared.**
>
> $\textbf{Response 1:}$
> Thanks for your valuable comments. We compare with Ghosh et al [1] in Table 15 and analyze the results. Concretely, the simple but effective Visual Description Grounded Decoding (VDGD) [1] enhances visual perception and improves reasoning capabilities by first generating a detailed description of the image and appending it as a prefix to the instruction. We compare VDGD with our SID in terms of hallucination evaluation benchmarks( i.e., POPE and CHAIR (testing on the same 500 images)) and the general ability benchmark (i.e., MMbench) and add detailed analyses. We re-implement VDGD [1] (version: arXiv:2405.15683v2) based on official codes on LLaVA-1.5 7B.
>
>
>
> |     | POPE (random)↑ | POPE (adversarial)↑ | CHAIRs↓  | CHAIRi↓ | MMbench↑ |
> | :---: | :---: | :---: | :---: | :---: | :---: |
> | Greedy | 88.8 | 79.1 |  49.6 | 14.4 | 64.4 |
> | OPERA | 88.9 | 79.2 |  45.2 | 12.7 | 64.4 |
> | VDGD[1]* | 89.0 | 79.4 | 46.7 | 13.7 | 65.2 |
> | SID* | 89.3 |83.3 | 44.2 | 12.2 | 65.0 |
>
>
> * denotes employing the greedy decoding strategy
>
>
>
> The results above demonstrate the effectiveness of VDGD [1] in alleviating hallucinations and enhancing general reasoning abilities in LVLMs.
> However, while the grounding visual descriptions generated by the LVLMs themselves effectively improve visual perception and reasoning capabilities, they may inevitably contain hallucinations.
> Consequently, VDGD is inferior in the POPE ($\textbf{adversial}$) subset, which prioritizes $\textbf{co-occurring}$ objects that are not present in the image. Meanwhile, VDGD shares somewhat similar motivations in enhancing vision information via Equation (6) as we analyzed. The experiments in Table 2 are consistent with the above results, indicating that boosting the vision information is effective in mitigating hallucinations but is less effective in adversarial environments.
>
>
> We add experimental results and analysis of VDGD in Table 15 of Appendix A.7, marked in blue, as follows:
>
> > **Line 1086**: Visual Enhancing Decoding Strategy. Although LVLMs can accurately recognize visual elements,....
>
>
>
>
> [1] Visual Description Grounding Reduces Hallucinations and Boosts Reasoning in LVLMs, Ghosh et al., 2024, arXiv:2405.15683v2.

---

> ### Author Response · Authors · 2024-11-21
> **Response to the Reviewer yTiw [Concerns 2]**
>
> $\textbf{Concern 2:}$ **A few more important benchmark comparisons are needed - MathVista, MMVet, LLaVA-Bench, MMMU.**
>
> $\textbf{Response 2:}$
> Thanks for your valuable suggestions.
> We conduct experiments on MathVista (testmini), MMVet, LLaVA-Bench, and MMMU, which mainly focus on the general and complex reasoning abilities of LVLMs, to validate the effectiveness of SID. Due to time constraints, we do our best to compare SID with two representative decoding methods (i.e., VCD and OPERA) on LLaVA-1.5 7B.
>
>
>
> |     | Greedy | VCD* | OPERA | SID* |
> | :---: | :---: | :---: | :---: | :---: |
> | MathVista [1] | 27.1 | 26.3 | 27.1 | **27.4** |
> | MMVet [2] | 31.1 | 30.2 | 31.1 | **31.2** |
> | LLaVA-Bench [3] | 63.4 | 63.6 | 64.3 | **68.7** |
> | MMMU [4]| 32.6 | 31.0 | 32.6 | **33.4** |
>
>
> $*$ denotes employing the greedy decoding strategy
>
>
> The above table indicates that our SID improves the reasoning ability, especially in the LLaVA-Bench benchmark. However, VCD slightly degrades LVLM's complex reasoning ability, as reflected in MathVista, MMVet, and MMMU benchmarks. OPERA has little gains in terms of discrimination tasks. These results are consistent with Table 5 (MME and MMbench benchmarks), Table 3, and Table 4.
>
> Note that our paper has been validated on various benchmarks and metrics, including POPE (Table 4), CHAIR (Table 3), GPT-4 assisted Sentence-level Hallucination Ratio (SHR) (Figure 7), 1/2-gram, the number of generated words/sentences per image (WPI/SPI) (Figure 7), GPT4-V assisted evaluation (Table 7), MME, and MMBench (Table 5) to evaluate the generated texts regarding $\textbf{hallucination}$, $\textbf{fluency}$, $\textbf{detailness}$, and various $\textbf{general}$ ability.
>
>
> We add experimental results of MathVista (testmini), MMVet, LLaVA-Bench, and MMMU  in Table 8 of Appendix A.5.
>
> > **Line 905**: APPENDIX A.5 MORE GENERAL BENCHMARKS EVALUATION...
>
>
>
>
> [1] MathVista: Evaluating Mathematical Reasoning of Foundation Models in Visual Contexts, ICLR 2024.
>
> [2] MM-Vet: Evaluating Large Multimodal Models for Integrated Capabilities, ICML 2024.
>
> [3] Visual instruction tuning, NeurIPS 2023.
>
> [4] Mmmu: A massive multi-discipline multimodal understanding and reasoning benchmark for expert agi, CVPR 2024.
>
>
> Hope these baseline comparisons and experiments could address your concerns. Please let me know if you have any concerns and we will be more than happy to answer them.
>
> Best,
>
> Authors

---

> ### Comment · Reviewer_yTiw · 2024-11-24
> **Response to the authors**
>
> Thank you for clarifying all my queries. I have increased my score from 5 to 6.

---

> > ### Author Response · Authors · 2024-11-24
> > **Thanks for your positive recommendation**
> >
> > Dear Reviewer yTiw,
> >
> > Thank you for taking the time to review our paper. We sincerely appreciate your thoughtful feedback and positive recommendation!
> >
> > Best,
> >
> > Authors

---

### Official Review · Reviewer_d52A · 2024-11-03

**Soundness:** 3
**Presentation:** 3
**Contribution:** 3
**Rating:** 6
**Confidence:** 4

**Summary:**

This paper proposes Self-Introspective Decoding (SID) for mitigating hallucinations in Large Vision-Language Models (LVLMs). The authors specifically observe that LVLMs are capable of introspectively assessing the importance of vision tokens based on preceding tokens. Based on this, the authors propose a Context and Text-aware Token Selection ($\text{CT}^2\text{S}$) strategy to amplify fine-grained hallucinations, which are ultimately mitigated using a contrastive decoding method. Comprehensive experimental results across various benchmarks demonstrate that the proposed SID outperforms other contrastive decoding baselines in alleviating hallucinations in LVLMs, while also achieving lower computational cost.

**Strengths:**

1. The proposed method seems novel and interesting. The motivation is also clear.
2. Through empirical experiments, the authors highlight that current vision-and-text agnostic contrastive decoding methods can introduce uncertainty noise and degrade performance. This provides valuable insights for future research.
3. The experimental results are comprehensive and promising, effectively validating the effectiveness of the proposed approach. Efficiency comparisons further highlight the proposed method’s advantage over other contrastive decoding approaches.
4. The authors provide the code implementation for this work, enhancing its reproducibility (though I have not run the code myself).







5. The writing and presentation of this paper is good.

**Weaknesses:**

1. The proposed method shares some similarities with AVISC [1], which also identifies less-important visual tokens for contrastive decoding. A detailed discussion of these similarities, along with a comparison of experimental results, would strengthen this paper.
2. In Figure 6, pruning all vision tokens results in less than a 0.5% performance difference from the optimal setting, suggesting that the proposed strategy of selecting the top-k least important tokens provides only marginal performance gains. Additionally, could you clarify the statement, “*The loss of vision information for subsequent decoder layers results in losing the visual context, leading to aimless hallucinations without sufficient vision grounding*”? Are there any examples or experiments that illustrate this?
3. The proposed method achieves marginal performance improvements on the MME and MMBench benchmarks. Could you please include the standard deviations for the two benchmarks, as their performance is sensitive to random seeds?
4. The proposed method may lack interpretability, as the selected less important tokens (shown in Figures 3 and 4) do not carry true semantic meaning, unlike other contrastive decoding approaches such as HALC [2]. Besides, the proposed method seems not very reasonable when applied to simple LVLMs such as InstructBLIP, which only has 32 vision tokens. In this case, preserving only the least important tokens provides a very coarse measure of vision importance and is unlikely to effectively induce vision-and-text association hallucinations.

Minor issues:
1. In Table 6, is the inference time calculated per instance or for the entire benchmark?
2. Wrong citation format in Line 910.
3. In Table 1, why are there no standard deviations reported for the VCD method in the Sampling setting? Additionally, are the three experiments conducted using three different random seeds? The reported deviations are lower than expected.

[1] Woo, S., Kim, D., Jang, J., Choi, Y., & Kim, C. (2024). Don't Miss the Forest for the Trees: Attentional Vision Calibration for Large Vision Language Models. *arXiv preprint arXiv:2405.17820*.

[2] Chen, Z., Zhao, Z., Luo, H., Yao, H., Li, B., & Zhou, J (2024). HALC: Object Hallucination Reduction via Adaptive Focal-Contrast Decoding. In *International Conference on Machine Learning*.

**Questions:**

1. In Table 1, the authors demonstrate that removing the adaptive plausibility constraint resulted in lower performance degradation for the proposed method on the POPE benchmark. Does this also hold for the open-ended CHAIR benchmark and the more comprehensive MME benchmark?
2. How do you define vision-and-text association hallucination?
3. What is the specific experimental setup in Figure 5? Why does ID produce entirely inconsistent responses?
4. How do the proposed method and other baselines perform on the Existence, Count, Position, and Color subsets of the MME benchmark?
5. The experiment in Lines 346-362 is interesting. However, the rationale behind why adding underperforms the proposed method is unclear. From my understanding, boosting the target logits of the original prediction could also enhance discrimination.

---

> ### Author Response · Authors · 2024-11-21
> **Response to the Reviewer d52A [Concerns 1-2]**
>
> Thank you for the insightful reviews!
> We are thrilled that our paper is considered by the reviewer as having the novel and interesting methodology, valuable insights, comprehensive experiments and low computational cost.
> We appreciate the opportunity to clarify details of our motivations, statements, and experimental results. Here, we answer the comments point by point.
>
> $\textbf{Concern 1:}$ **Comparisons and Discussion with AVISC [1].**
>
> $\textbf{Response 1:}$ Thank you for the constructive suggestion. We want to clarify in terms of motivation, robustness, and experiments.
>
>
> $\textbf{Motivation}$: AVISC [1], in contrast to ours, preserves outlier high attention tokens (named 'blind token') and substracts output logits to counteract the over-emphasis of 'blind token'. In this way, AVISC promotes balanced consideration of all tokens to alleviate hallucinations [1]. Differently, SID preserves spurious related tokens to amplify fine-grained hallucinations and mitigate them.
>
>
> $\textbf{Robustness}$: Similar to 'artifact token' [2], 'blind token' aggregates global information and maintains global discrimination ability. Appendix Table 9 shows that the Top-100 vision tokens with high attention scores can largely maintain the original performance. Therefore, 'blind token' tends to have $\textbf{high probability}$ of target class logits, and contrastive decoding does not improve the target class's probability while might bring extra noise. AVISC heavily $\textbf{relies on}$ the adaptive plausibility constraint (Eq. 3) and often degrades the original greedy decoding performance, similar to VCD and ICD, as indicated by the experiments below.
>
> $\textbf{Experiments}$: Due to time constraints, we compare AVISC on both greedy and sampling settings in POPE and CHAIR benchmarks with LLaVA-1.5 7B. Although AVISC outperforms vanilla sampling decoding under adaptive plausibility constraint Eq. (3), it still degrades the greedy decoding to some extent, which indicates  that the attentional vision calibration (AVISC) strategy induces some annoying noise.
>
>
>
> |     | POPE (random)↑ | POPE (popular)↑ |  POPE (adversarial)↑ |CHAIRs↓  | CHAIRi↓ |
> | :---: | :---: | :---: | :---: | :---: | :---: |
> | Sampling | 84.77 | 79.98 |  76.03 | 51.3 | 16.8 |
> | VCD^ | 86.84 | 82.65 | 77.31 | 48.0 |  13.9 |
> | AVISC^ | 87.41 | 83.46 | 78.35 | 46.6 | 12.5  |
> | SID^ | 88.91 |83.97 | 82.54 | 45.0 | 11.7 |
> | Greedy | 88.81 | 82.76 | 79.11 | 49.6 |  14.4 |
> | VCD* | 87.02 | 83.53 | 78.12 | 46.8 |  13.2 |
> | AVISC* | 88.16 | 84.14 | 78.83 | 45.3 | 14.7 |
> | SID* | 89.46 |85.13 | 83.24 | 44.2 |12.2 |
>
>
> $*$ and ^ denote the greedy and sampling decoding strategies
>
> We add the analyses and experimental results in Table 13 of Appendix A.7, marked in blue, as follows:
>
> > **Line 1050**: A APPENDIX A.7 Token Selection Strategies Analyses. ...Moreover, AVISC (Woo et al., 2024)...
>
>
> [1] Don't Miss the Forest for the Trees: Attentional Vision Calibration for Large Vision Language Models. arXiv preprint arXiv:2405.17820, 2024.
>
> [2] Vision Transformers Need Registers. ICLR 2024.
>
>
> $\textbf{Concern 2:}$ **Clarify Figure 6 and statement.**
>
> $\textbf{Response 2:}$ Thanks for your careful concerns.
>
> 1. Figure 6 illustrates analyses of varying *i* and preserved ratios of vision tokens. Note that when pruning all vision tokens, the remaining tokens (instructions) *still* pass through subsequent decoding layers. The multimodal knowledge absorbed in the early decoder layer induces contextual hallucinations during subsequent decoding.
> 2. The ideal induced hallucination distributions should be *target-co-occurring* while suppressing *target logits*. If we prune all vision tokens, the absorbed visual information diminishes during subsequent decoding.
> An extreme case is VIG [1], which directly ablates vision input. The induced distributions of VIG are noisy (i.e., aimless hallucinations) due to insufficient vision grounding. Therefore, we preserve vision tokens with low attention values to induce *target-co-occurring* hallucinations while suppressing *target logits*.
>
>
> [1] Multi-modal hallucination control by visual information grounding. CVPR 2024.
>
> We explain and rephrase the statement, marked in blue, as followings:
>
> > **Line 329**: ...aggressive pruning of vision tokens (i.e., 0%) after layer 𝑖 =1 is not optimal. As the ideal induced hallucination distributions are *target-co-occurring* but suppress *target logits*, the loss of visual information for subsequent decoding results in visual context diminishing, which can lead to aimless hallucinations due to insufficient grounding in visual information...

---

> ### Author Response · Authors · 2024-11-21
> **Response to the Reviewer d52A [Concerns 3-5]**
>
> $\textbf{Concern 3:}$ **Standard deviations of MME and MMbench.**
>
> $\textbf{Response 3:}$ We add the standard deviations using three different seeds (i.e., 41, 42, 43) in the updated paper. Please note that we use the $\textit{greedy}$ decoding strategy, and the standard deviations are small.
> Our goal is to mitigate the hallucination issue, and results on MME and MMBench benchmarks demonstrate SID greatly $\textit{preserves}$ and $\textit{improves}$ LVLMs' various general abilities.
>
>
>
> |         |   Greedy   |  Sampling  |    DoLa    |     ICD    |     VCD    | OPERA      | SID        |
> |:-------:|:----------:|:----------:|:----------:|:----------:|:----------:|------------|------------|
> |   MME ↑  | 1510.8±1.2 | 1471.5±5.6 | 1480.7±.13 | 1473.2±1.2 | 1488.5±0.8 | 1515.2±1.1 | 1520.4±0.9 |
> | MMbench↑|  64.4±.22  |  63.9±.81  |  63.7±.22  |  63.0±.24  |  63.8±.22  | 64.4±.13   | 65.0±.23   |
>
> Table5: LVLM benchmark evaluations. DoLa, ICD, VCD,and SID employ the same greedy decoding.
>
>
>
>
> $\textbf{Concern 4:}$ **Lack interpretability of token selections and InstructBLIP, compared to HALC [1].**
>
> $\textbf{Response 4:}$ Thanks for your insightful comments.
> HALC [1] provides a more intuitive and interpretable approach by locating fine-grained tokens within the original image for each generated token.
> However, it is essential to clarify that HALC relies on $\textit{extra}$ pre-trained networks like Grounding DINO [2] / OWLv2 [3] to identify vision regions and BLIP [4] for vision matching, while our method aims to avoid utilizing auxiliary analysis networks and instead focuses on introspectively eliminating hallucinations. In Sec. 7 Future Work, we consider training the external network to automatically determine optimal hyperparameters and also consider enhancing the interpretability by leveraging extra analysis networks. The discussions are marked in blue as follows:
>
> > **Line 533**: ...In addition, to enhance the interpretability of hallucination alleviations, we consider resorting to pre-trained analysis networks to intuitively locate spurious related vision regions....
>
>
> [1] HALC: Object Hallucination Reduction via Adaptive Focal-Contrast Decoding. ICML 2024.
>
> [2] Grounding dino: Marrying dino with grounded pre-training for open-set object detection. ECCV 2024.
>
> [3] Scaling open-vocabulary object detection. NeurIPS 2023.
>
> [4] Blip: Bootstrapping language-image pre-training for unified vision-language
>  understanding and generation. ICML 20222
>
>
>
>  $\textbf{Concern 5:}$ **Minor issues regarding inference time of Table 6, wrong citation format in Line 910, and concerns of deviations in Table 1.**
>
> $\textbf{Response 5:}$ Thanks for your careful concerns!
>
> 1. We calculate the inference time of the entire benchmark to avoid bias. We add the explanations in the revised paper.
>
> 2. We revise the wrong citations in the revised paper.
>
> 3. We directly adopt results of VCD from the original paper in the sampling setting and re-implement VCD in the greedy setting, which has small deviations; We conduct experiments using three random seeds (i.e., 41, 42, 43).

---

> ### Author Response · Authors · 2024-11-21
> **Response to the Reviewer d52A [Concerns 6-10]**
>
> $\textbf{Concern 6:}$ **Does the adaptive plausibility constraint affect CHAIR and MME?**
>
> $\textbf{Response 6:}$ Yes, the adaptive plausibility constraint also greatly affects CHAIR and MME. Due to space constraints, we do not show the results of CHAIR and MME in Table 1. In the updated paper, we add the experimental results of CHAIR and MME in the Appendix Table 11.
>
>
>
> |           | CHAIRs↓ | CHAIRi↓ | MME↑    |
> |:---------:|:------:|:------:|--------|
> | Sampling  | 51.3   | 16.8   | 1471.5 |
> |    ICD    |  48.7  |  13.9  | 1479.7 |
> | w/o Eq. 3 |  52.2  |  16.6  | 1448.5 |
> |    VCD    |  48.0  |  14.3  | 1481.9 |
> | w/o Eq. 3 |  51.9  |  17.2  | 1457.1 |
> |    SID    |  45.0  |  11.7  | 1487.3 |
> | w/o Eq. 3 | 46.5   | 12.2   | 1484.6 |
>
> All methods adopt the sampling decoding strategy, and the results are the average of three running times with three random seeds.
>
> We add the analysis of the adaptive plausibility constraint in Table 11 of Appendix A.7.
>
> > **Line 990**: APPENDIXA.7 More Analyses on Adaptive Plausibility Constraint...
>
>  $\textbf{Concern 7:}$ **How to define vision-and-text association hallucination?**
>
> $\textbf{Response 7:}$ 'Vision-and-text association' means 'vision-and-text aware', as opposed to being 'vision-and-text agnostic' (i.e., VCD and ICD). Vision and text are both fed to the LLM decoder, enabling multimodal information interactions. We focus on amplifying the vision-and-text association (or vision-and-text awareness) hallucinations informed by low-attention vision tokens.
>
>
>  $\textbf{Concern 8:}$ **The specific experimental setup of ID in Figure 5.**
>
> $\textbf{Response 8:}$ ID utilizes the negative prompt 'You are a confused image caption model.' following [1,2] as shown in the $\textit{upper right}$ of Figure 5. All other parameters are set to their default values. For the open-end generation task, LVLMs tend to follow the negative prompt, leading them to refuse to respond.
>
> [1] Mitigating Hallucinations in Large Vision-Language Models with Instruction Contrastive Decoding, ACL 2024 Findings.
>
> [2] Instructive Decoding: Instruction-tuned Large Language Models are Self-refiner from Noisy Instructions, ICLR 2024.
>
>
>  $\textbf{Concern 9:}$ **How do the performances on the subsets (i.e., Existence, Count, Position, and Color) of the MME benchmark?**
>
> $\textbf{Response 9:}$ Thanks for your insightful suggestion. Existence, Count, Position, and Color subsets of the MME benchmark cover both object-level and attribute-level hallucinations. Experimental results are illustrated below:
>
>
> |        | Existence | Count | Position | Color | Total |
> |:------:|:---------:|:-----:|:--------:|:-----:|:-----:|
> | Greedy |   182.3   | 130.3 |   126.8  | 155.7 | 594.1 |
> |  Dola*  |   180.1   | 127.4 |   119.3  | 154.4 | 581.2 |
> |   VCD*  |   179.5   | 128.1 |   123.8  | 155.5 | 586.9 |
> |  Ours*  |   183.9   | 132.2 |   127.8  | 155.9 | 599.8 |
>
>
> *denotes the greedy decoding strategies
>
> We add the results of Existence, Count, Position, and Color subsets of the MME benchmark in Table 9 of Appendix A.5.
>
> > **Line 945**: APPENDIX A.5...We also illustrate experimental results of Existence...
>
>
>  $\textbf{Concern 10:}$ **Explaining the interesting experimental results in Lines 346-362.**
>
> $\textbf{Response 10:}$ Thanks for appreciating our analysis! In the $\textit{random}$ setting, enhancing vision information typically boosts the target logits of the original prediction. However, in the $\textit{adversarial}$ setting, which is more challenging due to the emphasis on prioritizing co-occurring confusing objects, the enhanced vision information $\textit{might not}$ boost target logits but instead boost the logits of co-occurring objects. It is $\textit{easier}$ to prune important vision tokens to amplify the distribution of co-occurring objects and then subtract them.
>
>
> Hope these explanations could address your concerns. Please let me know if you have any concerns and we will be more than happy to answer them.
>
> Best,
>
> Authors

---

> > ### Comment · Reviewer_d52A · 2024-11-24
> >
> > Thank you for your detailed responses and clarifications, which have addressed most of my concerns. However, I still have some unresolved concerns regarding Concerns 3 and 4:
> >
> > [a] The proposed method achieves only marginal performance improvements on the MME benchmark, such as a +9.6 increase compared to Greedy and just +5.2 compared to OPERA. Additionally, the performance gains of the proposed SID on the Existence, Count, Position, and Color subsets also appear limited. Does this mean that the proposed method might be less effective in addressing cases beyond object existence hallucinations?
> >
> > [b] Could you provide additional visualization results, similar to Figures 3 and 4, but using the InstructBLIP model?
> >
> > Given the clear motivation and novelty of this work, I will maintain my positive rating of 6.

---

> > > ### Author Response · Authors · 2024-11-24
> > > **Thanks for your positive recommendation and follow-up comments**
> > >
> > > Dear Reviewer d52A,
> > >
> > > Thank you for your follow-up comments! We will add detailed responses and clarifications later.
> > >
> > > Best,
> > >
> > > Authors

---

> > > > ### Author Response · Authors · 2024-11-24
> > > > **Response to the Reviewer d52A**
> > > >
> > > > Thank you for the valuable follow-up comments!
> > > > We appreciate the opportunity to discuss these with you. Here, we answer the concerns point by point.
> > > >
> > > >
> > > > $\textbf{Concern 1:}$ **Concerns on the performance on MME and its subsets.**
> > > >
> > > >
> > > > $\textbf{Response 1:}$ Thanks for your thoughtful concern. $\textit{Without}$ the assistance of auxiliary analysis networks like Grounding DINO [1] and OWLv2 [2], we believe our method might demonstrate significant improvements in the various complex settings of MME.
> > > > Additionally, our method clearly outperforms other decoding-based methods in the greedy setting, whereas contrastive decoding-based methods tend to degrade the original performance.
> > > >
> > > > To further validate the superior performance of SID, we present the results in the $\textit{sampling}$ decoding setting, which ensures diverse and coherent generated texts and is the default in both closed-source and open-source LVLMs.
> > > > The table shows that our method achieves **+43.7** and **+37.7** compared to sampling and VCD on MME and achieves **+33.4** and **+18.0** compared to sampling and VCD on MME subsets.
> > > >
> > > > |        |    MME    | Existence | Count     | Position | Color | Total |
> > > > |:------:|:---------:|:---------:|:-----:    |:--------:|:-----:|:-----:|
> > > > |Sampling |1471.5±5.6|  171.7±6.2| 120.8±7.8 |112.6±4.8 | 151.0±3.9 | 556.1 |
> > > > |  Dola*  |1474.3±6.8|  173.2±5.8| 122.4±5.5 |115.2±6.2 | 152.8±5.4 | 563.6 |
> > > > |   VCD*  |1477.5±7.2|  174.4±5.9| 124.1±7.7 |119.4±6.9 | 153.6±6.8 | 571.5 |
> > > > |  Ours*  |1515.2±4.7|  180.5±5.7| 130.7±5.9 |123.8±6.1 | 154.5±5.0 | 589.5 |
> > > >
> > > >
> > > > $*$ denotes the $\textit{sampling}$ decoding strategy. Experiments are conducted on LLaVA-1.5 7B.
> > > >
> > > > [1] Grounding dino: Marrying dino with grounded pre-training for open-set object detection. ECCV 2024.
> > > >
> > > > [2] Scaling open-vocabulary object detection. NeurIPS 2023.
> > > >
> > > > $\textbf{Concern 2:}$ **Visualizing the vision tokens of InstructBLIP model.**
> > > >
> > > >
> > > > $\textbf{Response 2:}$ Thanks you for the insightful concern. InstructBLIP condenses vision tokens from 256 to 32 by Q-former and feeds them into the LLM decoder. As a result, we are unable to visualize the dynamic token pruning process like Figures 3 and 4.
> > > > While it's true that the 32 vision tokens offer a relatively coarse measure of vision importance compared to vision-language connector-based methods like Shikra, LLaVA-1.5, and LLaVA-NeXT, integrating InstructBLIP with SID aligns with our motivation of adaptively amplifying vision-and-text association hallucinations. Seamlessly integrating SID with SID makes noticeable improvements, as shown in Tables 3, 4, 7, and 14 as well as Figure 7.
> > > >
> > > >
> > > >
> > > >
> > > > Hope the discussions could address your concerns. Please let me know if you have any concerns, and we will be more than happy to answer them.
> > > >
> > > > Best,
> > > >
> > > > Authors

---

> ### Comment · Reviewer_d52A · 2024-11-24
>
> Thank you for the authors' detailed follow-up responses. Please include these new results and discussions into the revised paper. I will maintain my positive rating and look forward to seeing this work accepted. Good luck!

---

> > ### Author Response · Authors · 2024-11-24
> > **Thanks for your positive recommendation**
> >
> > Dear Reviewer d52A,
> >
> > The revised paper is updated. Thank you for taking the time to give follow-up comments. We sincerely appreciate your thoughtful feedback and positive recommendation!
> >
> > Best,
> >
> > Authors

---

### Official Review · Reviewer_Xv5a · 2024-11-03

**Soundness:** 2
**Presentation:** 3
**Contribution:** 2
**Rating:** 5
**Confidence:** 4

**Summary:**

The paper introduces Self-Introspective Decoding (SID) to alleviate hallucinations in large vision language models. Other available methods either introduce noise or lead to double inference costs. SID offers an alternative using a Context and Text-aware Token Selection (CT2S) strategy that selectively attenuates less important vision tokens in early decoder layers. This reduces irrelevant hallucinations in the generations. The approach requires minimal additional computational resources. The paper presents multiple empirical results to show that the method reduces hallucinations while preserving text quality.

**Strengths:**

- The proposed approach does not require additional computational resources
- GPT-4 assisted analyses were done to calculate Sentence-level Hallucination Ratio
- The paper is well written

**Weaknesses:**

- The paper does not present the results on different benchmarks to show the preservation of LVLM ability. LVLMs should be extensively tested on a variety of benchmarks that test different skills, like - MathVision and Mathvista for mathematical reasoning, MMMU for college-level knowledge on various subjects, MM-Vet/v2
- The paper does not cover a comprehensive set of baselines - Woodpecker (https://arxiv.org/abs/2310.16045), LRV (https://www.researchgate.net/publication/375596083), LURE (https://arxiv.org/pdf/2310.00754)
- "Hallucination, defined as the generation of irrelevant, factually incorrect, or meaningless text in a given context." The approach aims to tackle hallucinations in LVLMs as a whole but does not mention or tackle style hallucinations or biases introduced when LVLMs are instruction-tuned.

-Minor:
(typo) Figure 2 - Gnerated

#256 validate

**Questions:**

1. How does the performance of the proposed approach vary across different LVLM sizes? Do we have any results for larger and smalled LVLMs? (other than 7B)
2. Where does the proposed approach fail to mitigate hallucinations?

---

> ### Author Response · Authors · 2024-11-21
> **Response to the Reviewer Xv5a [Concerns 1]**
>
> Thank you for the thoughtful reviews.
> We appreciate the opportunity to clarify some misunderstandings and add more discussions and benchmarks evaluation. Here, we answer the concerns point by point.
>
>
> $\textbf{Concern 1:}$ **Different benchmarks to show the preservation of LVLM ability.**
>
>
> $\textbf{Response 1:}$
> Thank you for the valuable comments.
> Kindly note that in addition to hallucination-related benchmarks and metrics (POPE, CHAIR, SHR, GPT4-V assisted correctness evaluation), our paper had been validated on $\textit{various}$ benchmarks and metrics including 1/2-gram, the number of generated words/sentences per image (WPI/SPI) (in Figure 7), GPT4-V assisted detailedness evaluation (in Table 7), $\textbf{MME}$, and $\textbf{MMBench}$ (in Table 5) to evaluate the generated texts regarding $\textbf{fluency}$, $\textbf{detailness}$, and LVLMs' various $\textbf{general}$ abilities.
> MME comprises $\textbf{ten}$ sub-tasks to evaluate model’s perceptual capabilities and $\textbf{four}$ sub-tasks for assessing cognitive abilities. MMBench systematically evaluates $\textbf{twenty}$ ability dimensions of LVLMs.
> Experiments presented in $\textbf{Table 5}$ on MME and MMBench illustrate SID can maintain and improve various multimodal general abilities.
>
>
>
> To further analyze the abilities of LVLMs from different perspectives, we conduct experiments on **MathVista** (testmini), **MMVet**, **LLaVA-Bench**, and **MMMU**, which mainly focus on the general and complex reasoning abilities of LVLMs, to validate the effectiveness of SID. Due to time constraints, we do our best to compare SID with two representative decoding methods (i.e., VCD and OPERA) on LLaVA-1.5 7B.
>
>
>
> |     | Greedy | VCD* | OPERA | SID* |
> | :---: | :---: | :---: | :---: | :---: |
> | MathVista[1] ↑ | 27.1 | 26.3 | 27.1 | **27.4** |
> | MMVet [2] ↑ | 31.1 | 30.2 | 31.1 | **31.2** |
> | LLaVA-Bench [3] ↑ | 63.4 | 63.6 | 64.3 | **68.7** |
> | MMMU [4] ↑| 32.6 | 31.0 | 32.6 | **33.4** |
>
>
> $*$ denotes employing the greedy decoding strategy
>
>
> This table indicates that our SID enhances reason abilities, particularly in the LLaVA-Bench benchmark. However, VCD slightly degrades LVLM's complex reasoning ability, as reflected in MathVista, MMVet, and MMMU benchmarks. OPERA has little gains in discrimination tasks. These results are consistent with Table 5 (MME and MMbench benchmarks), Table 3, and Table 4.
>
> We add the experimental results for these benchmarks in Table 8 of Appendix A.5, highlighted in blue, as follows:
>
>
> > **Line 905**: APPENDIX A.5 MORE GENERAL BENCHMARKS EVALUATION...
>
>
> [1] MathVista: Evaluating Mathematical Reasoning of Foundation Models in Visual Contexts, ICLR 2024.
>
> [2] MM-Vet: Evaluating Large Multimodal Models for Integrated Capabilities, ICML 2024.
>
> [3] Visual instruction tuning, NeurIPS 2023.
>
> [4] Mmmu: A Massive Multi-discipline Multimodal Understanding and Reasoning Benchmark for Expert Agi, CVPR 2024.

---

> ### Author Response · Authors · 2024-11-21
> **Response to the Reviewer Xv5a [Concerns 2-4]**
>
> $\textbf{Concern 2:}$ **The paper does not cover a comprehensive set of baselines - Woodpecker, LRV, LURE.**
>
> $\textbf{Response 2:}$ Thank you for the attentive concern. It’s essential to clarify SID is a $\textbf{training-free}$ plug-and-play decoding strategy that does not requires $\textbf{robust}$ $\textbf{instruction}$ $\textbf{tuning}$ with $\textbf{curated datasets}$ (as in LRV and HA-DPO) and $\textbf{post-hoc}$ utilizing $\textbf{auxiliary analysis}$ $\textbf{networks}$ (as in Woodpecker and LURE).
> This distinction is highlighted in **bold** in Section 2, "Related Work" (**Lines 115-119**: describing three different types).
> Therefore, it might be $\textbf{unfair}$ to directly compare SID with Woodpecker, LURE, LRV, and HA-DPO.
>
> To further validate the effectiveness of SID, we compare SID with auxiliary analysis networks-based method (i.e., LURE [1]) and the robust instruction tuning-based method (i.e., HA-DPO [3]). We also apply the plug-and-play SID to LURE and HA-DPO on the POPE, CHAIR, and MME benchmarks following their official implementations.
>
>
>
> |        |  POPE↑ | CHAIRs↓ | CHAIRi↓ |   MME↑  |
> |:------:|:-----:|:------:|:------:|:------:|
> | Greedy |  78.6 |  55.4  |  14.2  |  805.7 |
> |  LURE  |  78.7 |  55.1  |  14.0  |  846.2 |
> |  +VCD  |  78.3 |  55.0  |  14.3  |  813.4 |
> |  +SID  |  82.5 |  52.1  |  13.6  |  854.9 |
> |   SID  |  82.2 |  51.9  |  13.2  |  838.1 |
> | Greedy | 83.6  |  49.6  |  14.4  | 1510.8 |
> | HA-DPO | 85.4  |  44.7  |  13.6  | 1522.3 |
> |  +VCD  | 83.0  |  46.1  |  13.8  | 1500.6 |
> |  +SID  | 86.2  |  43.8  |  12.4  | 1532.7 |
> |   SID  | 85.9  |  44.2  |  12.2  | 1525.3 |
>
>
> Note that all methods adopt the greedy decoding strategy. LURE and HA-DPO adopt MiniGPT-4 13B and LLaVA-1.5 7B, respectively.
>
> Experimental results indicate that SID outperforms LURE and HA-DPO in most cases. Additionally, seamlessly integrating SID with LURE and HA-DPO can significantly enhance performance.
>
> We add experimental results of more baselines in Table 10 of  Appendix A.7, as follows:
>
> > **Line 905** APPENDIX A.5 More Analyses on Other Baselines...
>
> [1] Analyzing and Mitigating Object Hallucination in Large Vision-language Models, ICLR 2024.
>
> [2] Aligning Large Multi-Modal Model with Robust Instruction Tuning, ICLR 2024.
>
> [3] Beyond Hallucinations: Enhancing Lvlms through Hallucination-aware Direct Preference Optimization, arXiv:2311.16839, 2024.
>
> $\textbf{Concern 3:}$ **The approach aims to tackle hallucinations in LVLMs as a whole but does not mention or tackle style hallucinations or biases introduced when LVLMs are instruction-tuned.**
>
> $\textbf{Response 3:}$ Thank you for the thoughtful concern!
>
> Our method proposes the $\textit{training-free}$ and $\textit{plug-and-play}$ decoding strategy to mitigate hallucinations. The self-introspective mechanism automatically amplifies the contextual hallucinations and then alleviates them.
>
> Although employing auxiliary analysis networks or instruction tuning with curated datasets can explicitly diagnose hallucination styles and biases, self-introspective decoding is free from external networks and instruction tuning and can also be seamlessly integrated with the above methods.
>
> In the updated paper, we add the detailed quantitative results and analysis of fine-grained hallucination types, specifically focusing on object-level (existence and count) and attribute-level (position and color) hallucination subsets in Appendix A.5 Table 9.
>
> $\textbf{Concern 4:}$ **Two typos.**
>
> $\textbf{Response 4:}$ Thank you for pointing typos out!  We update the figure and revise the typos.

---

> ### Author Response · Authors · 2024-11-21
> **Response to the Reviewer Xv5a [Concerns 5-6]**
>
> $\textbf{Concern 5:}$ **How does the performance of the proposed approach vary across different LVLM sizes?**
>
> $\textbf{Response 5:}$ Thanks for the thoughtful concern! We want to clarify that the SID has been validated across various scales ($\textbf{7B}$, $\textbf{8B}$, and $\textbf{13B}$) in Tables 4 and 14. Concretely, Table 4 illustrates the performance on LLaVA-NeXT, which utilizes LLaMA3 8B. Additionally, Table 14 (previously Table 10) in Appendix A.7 $\textbf{Larger-scale}$ $\textbf{LVLM}$ $\textbf{Backbone}$ shows the performance on LLaVA-1.5 13B and InstructBLIP 13B.
>
>
> $\textbf{Concern 6:}$ **Where does the proposed approach fail to mitigate hallucinations?**
>
> $\textbf{Response 6:}$ Thanks for raising this concern. Since our approach is a training-free decoding method that does not rely on auxiliary analysis networks, it inherently carries over the existing weaknesses of LVLMs. Intuitive case studies, as illustrated in Appendix Figures 13, 14, and 15, reveal that SID still generates some hallucinations, particularly in finer details such as eye color and vehicle identification specifics.
>
> These issues may arise from the relatively limited visual perception capabilities of the vision encoder. For future work, it is promising to integrate SID with InternVL [1], which scales the vision encoder up to 6B,  or consider leveraging auxiliary analysis networks like Grounding DINO [2] or OWLv2 [3] to address the internal weaknesses of LVLMs.
>
> We include failure analysis in Appendix A.6 CASE STUDY, marked in blue:
>
> > **Line 955** APPENDIX A.6 CASE STUDY. As we propose...
>
>
> [1] InternVL: Scaling up Vision Foundation Models and Aligning for Generic Visual-Linguistic Tasks, CVPR 2024.
>
> [2] Grounding dino: Marrying dino with grounded pre-training for open-set object detection. ECCV 2024.
>
> [3] Scaling open-vocabulary object detection. NeurIPS 2023.
>
> Hope these explanations and experimental results could address your concerns. Please let me know if you have any concerns and we will be more than happy to answer them.
>
> Best,
>
> Authors

---

> > ### Comment · Reviewer_Xv5a · 2024-11-25
> > **Response to Rebuttal**
> >
> > Thank you for taking the time to answer my questions in detail. I am satisfied with the authors' answers. However, I feel that the paper, as it stands now, is heavily reliant on the appendix. Important experiments and discussions must be included in the paper to create a better storyline and overall presentation of the proposed approach. Hence, I have updated my score accordingly.

---

> > > ### Author Response · Authors · 2024-11-25
> > > **Thanks for your satisfaction with our answers**
> > >
> > > Dear Reviewer Xv5a,
> > >
> > > Thank you for the thoughtful follow-up comments!
> > > We are glad that the rebuttal has addressed your concerns.
> > >
> > > Regarding the article structure, we would like to gently remind you that we have added the hyperlinks to the appendix and also highlighted them in bold for clarity.
> > >
> > > To better present the proposed approach, we add more discussions in the main body and provide clear hyperlinks to detailed results and analysis in the appendix, as shown in the revised paper.
> > >
> > >
> > > Best,
> > >
> > > Authors

---

> > > > ### Author Response · Authors · 2024-12-03
> > > > **Official Comment by Authors**
> > > >
> > > > Dear Reviewer Xv5a,
> > > >
> > > > Thank you very much for taking the time to review our paper.
> > > >
> > > > We would like to kindly remind you that the revised paper has been updated. It now includes more discussions in the main body and provides clear hyperlinks to detailed results and analyses in the appendix.
> > > >
> > > > Here are several key revisions:
> > > >
> > > > + Line 370 further explains why we compare decoding-based methods rather than tuning-based and auxiliary network-based methods;
> > > >
> > > > + Lines 368 and 490 highlight and link to the larger-scale backbone results in Appendix A.7 twice times;
> > > >
> > > > + Lines 380-387 describe the comprehensive experiments and link to the more general benchmark evaluations in Appendix A.6.
> > > >
> > > > In this context, we kindly and respectfully ask you to consider re-evaluating our paper’s presentation and score rating.
> > > >
> > > > Best regards,
> > > >
> > > > Authors

---

### Official Review · Reviewer_LdUB · 2024-11-03

**Soundness:** 3
**Presentation:** 3
**Contribution:** 3
**Rating:** 8
**Confidence:** 3

**Summary:**

This manuscript proposes to solve the hallucination issue in Large Vision-Language Models (LVLMs). The proposed method named Self-Introspective Decoding (SID) aims to solve the issue with a different decoding strategy compared to existing ones. The Context and Text-aware Token Selection (CT2S) strategy within SID preserves only unimportant vision tokens after the early layers of LVLMs, in order to amplify text-informed hallucinations during the auto-regressive decoding process and guide the LVLMs to produce more accurate outputs. Evaluation was conducted using four representative LVLMs: InstructBLIP, Shikra, LLaVA-1.5, and LLaVA-NeXT. Evaluation metrics include CHAIR, POPE, GPT-4 Assisted Evaluations and MME and MMBench Evaluations. Performance of the proposed SID was compared with Sampling, Greedy, Dola, and LVLM decoding strategies (VCD, ICD, and OPERA). Experiment results demonstrate the effectiveness of SID to generates less-hallucination and higher-quality texts, with lower additional computation cost.

**Strengths:**

1. The proposed SID with its key component: Context and Text-aware Token Selection (CT2S) strategy is a Training-Free decoding strategy which efficiently and effectively solves hallucination problem in LVLMs without additional costs.
2. Extensive experiments were conducted and several evaluation metrics were applied to verify the effectiveness of the SID in various aspects and the superiority over existing methods.
3. Hyperparameter sensitivity evaluation showed the proposed SID’s robustness to different hyperparameter settings.

**Weaknesses:**

1. The proposed SID is performed on pre-trained LVLMs, so it’s possible that the performance of SID is limited by those pre-trained models.
2. And currently there’s no solution to specifically tune SID for different LVLMs.

**Questions:**

Major comments:
1. Table 1 is not explained clear enough, according to the results whatever methods used, greedy decoding setting always performed better, than what is the significance of considering the constraints in Equ 3?
2. What about the possibility of integrating CT2S with other hallucination alleviation strategies?

Minor comments:
1. Better to keep names of the metrics consistent, e.g., CHAIRI and CHAIRS might be written as CHAIRi and CHAIRs, to be the same as Table 8-9.

---

> ### Author Response · Authors · 2024-11-21
> **Response to the Reviewer LdUB [Concerns 1-3]**
>
> Thank you for the insightful reviews!
> We appreciate the opportunity to add more explanations and discussions of the proposed method. Below, we address the concerns point by point.
>
>
> $\textbf{Concern 1:}$ **The proposed SID is performed on pre-trained LVLMs, so it’s possible that the performance of SID is limited by those pre-trained models.**
>
> $\textbf{Response 1:}$ Thanks for your insightful concern.
> Since our method is a training-free, plug-and-play decoding approach, it does indeed inherit the existing limitations of LVLMs.
> As demonstrated in the case studies in Appendix Figures 13, 14, and 15, SID still generates some hallucinations, particularly in finer details such as eye color and vehicle identification specifics.
> These issues may stem from the relatively limited visual perception capabilities of the vision encoder. For future work, it is promising to integrate SID with InternVL [1], which scales the vision encoder up to 6B,  or consider leveraging auxiliary analysis networks like Grounding DINO [2] or OWLv2 [3] to mitigate LVLMs' internal weaknesses.
>
>
> We add an analysis of SID faults, which inherit weaknesses from the pre-trained LVLM, in Appendix A.6 under CASE STUDY. These sections are highlighted in blue for clarity.
>
>
> [1] InternVL: Scaling up Vision Foundation Models and Aligning for Generic Visual-Linguistic Tasks, CVPR 2024.
>
> [2] Grounding dino: Marrying dino with grounded pre-training for open-set object detection. ECCV 2024.
>
> [3] Scaling open-vocabulary object detection. NeurIPS 2023.
>
> $\textbf{Concern 2:}$ **Currently, there’s no solution to specifically tune SID for different LVLMs.**
>
> $\textbf{Response 2:}$ Thank you for the careful concern.  Since SID is designed to be parameter-free, we have not proposed tuning-based methods. However, in $\textbf{Section 7}$, we outline two potential future directions: $\textbf{1)}$ As the pruning ratios and layer are set manually, we consider training the external network to automatically determine optimal hyperparameters, inspired by [1]. In addition, to enhance the interpretability of hallucination alleviations, we consider resorting to pre-trained analysis networks to intuitively locate spurious related vision regions. $\textbf{2)}$ Moreover, given that SID amplifies fine-grained hallucinations, we consider leveraging the CT$^2$S strategy to automatically construct high-quality negative instruction for robust visual instruction tuning rather than relying on expensive GPT-4 [2,3]. Note that the self-generated hallucination dataset ensures $\textit{style consistency}$, which is crucial for preference learning [3].
>
>
> [1] Diffrate: Differentiable Compression Rate for Efficient Vision Transformers, CVPR 2024.
>
> [2] Mitigating Hallucination in Large Multi-Modal Models via Robust Instruction Tuning, ICLR 2024.
>
> [3] Beyond Hallucinations: Enhancing Lvlms through Hallucination-aware Direct Preference Optimization, arXiv:2311.16839, 2024
>
> $\textbf{Concern 3:}$ **Explain the results of greedy and sampling decoding in Table 1.**
>
> $\textbf{Response 3:}$ Thanks for your thoughtful comment. Greedy decoding consistently outperforms sampling decoding regarding hallucination metrics. $\textit{However}$, greedy decoding generates repetitive text and often only reaches a local optimum [1,2]. In contrast, sampling decoding generates diverse and coherent output texts [2,3]. As a result, most open-source and closed-source LVLMs utilize the sampling decoding by default to facilitate diverse and coherent chat interactions.
> It is therefore practically meaningful to consider the constraints in Eq. 3 when analyzing sampling decoding.
>
> More detailed introductions of decoding strategies are in APPENDIX A.1 MORE BACKGROUNDS Decoding Strategy in LLMs. We add further explanations in the revised paper in APPENDIX A.1, highlighted in blue.
>
> [1] Neural text generation with unlikelihood training, ICLR 2020.
>
> [2] On decoding strategies for neural text generators, Transactions of the Association for Computational Linguistics, 2022.
>
> [3] The curious case of neural text degeneration, ICLR,2020.

---

> > ### Author Response · Authors · 2024-11-21
> > **Response to the Reviewer LdUB [Concerns 4-5]**
> >
> > $\textbf{Concern 4:}$ **What about the possibility of integrating CT2S with other hallucination alleviation strategies?**
> >
> > $\textbf{Response 4:}$ Thanks for the insightful seggestion.
> > 1) In Section 7 CONCLUSION AND FUTURE WORK, we consider leveraging the CT$^2$S strategy to automatically construct high-quality negative instruction for robust visual instruction tuning rather than relying on expensive GPT-4 [1,2]. Note that the self-generated hallucination dataset ensures $\textit{style consistency}$, which is crucial for preference learning [2].
> > 2) To further validate the effectiveness of SID, we compare SID with the auxiliary analysis networks-based method (i.e., LURE [1]) and the robust instruction tuning-based method (i.e., HA-DPO [3]) and apply the plug-and-play SID to LURE and HA-DPO on POPE, CHAIR, and MME benchmarks, following their official implementations.
> >
> >
> > |        |  POPE↑ | CHAIRs↓ | CHAIRi↓ |   MME↑  |
> > |:------:|:-----:|:------:|:------:|:------:|
> > | Greedy |  78.6 |  55.4  |  14.2  |  805.7 |
> > |  LURE  |  78.7 |  55.1  |  14.0  |  846.2 |
> > |  +VCD  |  78.3 |  55.0  |  14.3  |  813.4 |
> > |  +SID  |  82.5 |  52.1  |  13.6  |  854.9 |
> > |   SID  |  82.2 |  51.9  |  13.2  |  838.1 |
> > | Greedy | 83.6  |  49.6  |  14.4  | 1510.8 |
> > | HA-DPO | 85.4  |  44.7  |  13.6  | 1522.3 |
> > |  +VCD  | 83.0  |  46.1  |  13.8  | 1500.6 |
> > |  +SID  | 86.2  |  43.8  |  12.4  | 1532.7 |
> > |   SID  | 85.9  |  44.2  |  12.2  | 1525.3 |
> >
> >
> > Note that all methods adopt the greedy decoding strategy. LURE and HA-DPO adopt MiniGPT-4 13B and LLaVA-1.5 7B architectures, respectively.
> >
> > Experimental results indicate that SID outperforms LURE and HA-DPO in most cases. Additionally, seamlessly integrating SID with LURE and HA-DPO can significantly enhance performance.
> >
> > We add experimental results of more baselines in Table 10 of Appendix A.7 as follows:
> >
> > > **Line 964**: APPENDIX A.5 More Analyses on Other Baselines...
> >
> >
> > [1] Mitigating Hallucination in Large Multi-Modal Models via Robust Instruction Tuning, ICLR 2024.
> >
> > [2] Beyond Hallucinations: Enhancing Lvlms through Hallucination-aware Direct Preference Optimization, arXiv:2311.16839, 2024
> >
> > $\textbf{Concern 5:}$ **Keep names of the metrics consistent**.
> >
> > $\textbf{Response 5:}$ Thank you for the attentive suggestion! We revise CHAIRS and CHAIRI to CHAIRs and CHAIRi, ensuring consistency throughout the entire article.
> >
> > Hope these explanations and discussions could address your concerns. Please let me know if you have any concerns, and we will be more than happy to answer them.
> >
> > Best,
> >
> > Authors

---

> > > ### Comment · Reviewer_LdUB · 2024-11-26
> > >
> > > Thank you for answering my questions accordingly, which solves my concerns.

---

> > > > ### Author Response · Authors · 2024-11-26
> > > > **Thanks for your positive recommendation**
> > > >
> > > > Dear Reviewer LdUB,
> > > >
> > > > Thanks for taking your time to to review our paper! We sincerely appreciate your thoughtful concerns and positive recommendation.
> > > >
> > > > Best,
> > > >
> > > > Authors

---

### Author Response · Authors · 2024-11-22
**General Response**

Dear Reviewers:

We sincerely thank all the reviewers for their careful review and constructive suggestions, which have led to significant improvements in the revised manuscript.
We are thrilled that our paper is recognized by the reviewers for its novel and effective methodology (**LdUB**, **d52A**), valuable insights and motivations (**d52A**, **yTiw**), comprehensive experiments (**LdUB**, **d52A**, **yTiw**), low computational costs (**all reviewers**), and clear presentation (**all reviewers**).

We have carefully revised our manuscript based on reviewers' thoughtful feedback. All changes in the revised manuscript are marked in blue. In addition to the point-by-point responses, we have summarized the revisions as follows:

- We evaluate on more general benchmarks (i.e.,  MathVista, MMVet, LLaVA-Bench, MMMU) as shown in Table 8 of Appendix A.5.
- We illustrate the results of MME hallucination subsets  (i.e.,  Existence, Count, Position, Color) in Table 9 of Appendix A.5.
- We compare tuning-based and auxiliary networks-based baselines and deploy the proposed method into these baselines to validate its effectiveness in Table 10 of Appendix A.7.
- We further validate the impact of the 'adaptive
plausibility constraint' (Eq. 3) by utilizing CHAIR and MME benchmarks in Table 11 of Appendix A.7.
- We add comparisons with suggested baselines in Tables 13 and 15 of Appendix A.5 and provide detailed discussions.
- We add more explanations about Figure 6, Table 6, results deviation, failure cases, and future works. We carefully revise the typos.

Thank you once again for your invaluable comments. If you have additional comments or concerns, please let us know and we will be more than happy to answer.

Best,

Authors

---

### Meta-Review · Area_Chair_nnrW · 2024-12-20

**Metareview:**

This work brings a new method, named Self-Introspective Decoding (SID), that alleviates hallucinations in large vision language models. Existing approaches either introduce noise or need double inference costs, while this paper provides an alternative that uses a Context and Text-aware Token Selection scheme to selectively attenuate less important tokens in early decoder layers, and thus mitigates irrelevant hallucinations. The designed SID and the Context and Text-aware Token Selection scheme is a Training-Free strategy that effectively handles the hallucination issue without additional costs. All reviewers acknowledge the contributions of this work. In the camera ready version, authors still need to carefully improve the work following reviewers' comments.

**Additional Comments On Reviewer Discussion:**

Reviewers requested additional experiments and details. Authors have successfully addressed most of them. One reviewer mentioned that many important contents in the Supp need to be moved to the main paper.

---

### Decision · Program_Chairs · 2025-01-22

Accept (Poster)